# Assessing biomass and primary production of microphytobenthos in depositional coastal systems using spectral information

**Pascalle Jacobs**[1]*, **Jaime Pitarch**[1¤], **Jacco C. Kromkamp**[2†], **Catharina J. M. Philippart**[1]

**1** Department of Coastal Systems, Royal Netherlands Institute for Sea Research, Den Burg, the Netherlands,
**2** Department of Estuarine and Delta Systems, Royal Netherlands Institute for Sea Research, Yerseke, the Netherlands

† Deceased.
¤ Current address: Istituto di Scienze Marine, Consiglio Nazionale delle Ricerche, Roma, Italy
* pascalle.jacobs@nioz.nl

**Data Availability Statement:** After the manuscript is accepted for publication the authors will place the data used for drafting this manuscript in the data repository https://data.4tu.nl/. The data can

## Abstract

In depositional intertidal coastal systems, primary production is dominated by benthic microalgae (microphytobenthos) inhabiting the mudflats. This benthic productivity is supporting secondary production and supplying important services to humans including food provisioning. Increased frequencies of extreme events in weather (such as heatwaves, storm surges and cloudbursts) are expected to strongly impact the spatiotemporal dynamics of the microphytobenthos and subsequently their contribution to coastal food webs. Within north-western Europe, the years 2018 and 2019 were characterized by record-breaking summer temperatures and accompanying droughts. Field-calibrated satellite data (Sentinel 2) were used to quantify the seasonal dynamics of microphytobenthos biomass and production at an unprecedented spatial and temporal resolution during these years. We demonstrate that the Normalized Difference Vegetation Index (NDVI) should be used with caution in depositional coastal intertidal systems, because it may reflect import of remains of allochthonous pelagic productivity rather than local benthic biomass. We show that the reduction in summer biomass of the benthic microalgae cannot be explained by grazing but was most probably due to the high temperatures. The fivefold increase in salinity from January to September 2018, resulting from reduced river run-off during this exceptionally dry year, cannot have been without consequences for the vitality of the microphytobenthos community and its resistance to wind stress and cloud bursts. Comparison to historical information revealed that primary productivity of microphytobenthos may vary at least fivefold due to variations in environmental conditions. Therefore, ongoing changes in environmental conditions and especially extreme events because of climate change will not only lead to changes in spatiotemporal patterns of benthic primary production but also to changes in biodiversity of life under water and ecosystem services including food supply. Satellite MPB data allows for adequate choices in selecting coastal biodiversity conservation and coastal food supply.

then be accessed through DOI 10.4121/13516259, which has already been reserved for this purpose.

**Funding:** Financial support for data collection was provided to prof. Catharina J.M. Philippart by the Ministry of Infrastructure and Water Management, the Netherlands (grant number 31144695), the authors declare that the Ministry did not have any involvement in the study design, data collection & analysis or interpretation of the data. The ministry had not played a role in the decision to publish and was not involved in the preparation of the manuscript.

**Competing interests:** The authors have declared that no competing interests exist.

# Introduction

Coastal wetlands are highly productive marine systems, supporting high rates of secondary production and providing food for higher trophic levels as well as supplying important services to humans [1, 2]. But, worldwide, these systems are under intense pressure from coastal development, coastal erosion, dredging activities and reduced sediment fluxes from rivers [2, and references herein]. Other emergent threats to these systems come from changes in salinity, temperature and sea-level linked to climate change. This major concern with respect to threats to the natural values and provisioning services of coastal areas is explicitly addressed by the United Nations, in their sustainable development goals the ambition is to "avoid significant adverse impacts, including by strengthening their resilience, and take action for their restoration" and to "conserve at least 10 per cent of coastal and marine areas" [3, 4].

Coastal wetlands include tidal flats, sand, rock, or mud flats that undergo regular tidal inundation, which occupy more than 125,000 km$^2$ worldwide [2]. In intertidal deposits, tidal flats harbour microphytobenthos (MPB), which fuels coastal food webs [5]. These benthic microalgae are food for a variety of animals that graze on the algae directly from the sediment surface or filter them out of the water when resuspended by currents and wind-driven waves [6]. For marine UNESCO World Heritage sites, for example, high primary productivity by microphytobenthos supporting high numbers of migratory birds is considered as an outstanding unique value of coastal wetlands [7]. Primary production by benthic microalgae is especially important in tidal areas with a high proportion of mudflats that emerge during low tide [8–10] and in coastal areas that are characterised by turbid waters where production by pelagic microalgae is relatively low [11].

In most estuaries, water is too turbid for photosynthesis of benthic algae to occur during immersion [12, 13], and photosynthesis only takes place during emersion. Spatial variation in MPB within an estuary is shaped by the interlinked factors of tidal exposure and sediment type [6, 14, 15]. More elevated patches are exposed to daylight for a longer period, increasing the photosynthetic period. These areas are also characterised by low tidal energy, resulting in the deposition of finer sediments (silt, mud). Many studies reported a higher biomass of MPB in sediments with a high percentage of fine particles (mud) compared to more sandy sediments [16–20]. The main mechanisms behind the difference are thought to be the lower resuspension of algal cells in muddy habitats compared to sandy areas and higher concentration of nutrients in pore water of finer sediment [18, 21].

In temperate coastal systems, highest MPB biomass is found in spring, lowest biomasses in winter and in some areas a summer dip is observed [19, 22, 23]. The variation in biomass of microphytobenthos (MPB) is positively correlated with that in irradiance, temperature, and nutrient availability, while grazing, bioturbation and bacterial breakdown tend to decrease biomass [19]. Within estuaries, river discharges do not only influence salinity but also estuarine circulation [24]. Salinity changes result in species replacement [25] which might affect biomass. Resuspension due to wind and wave action causes generally a decrease in biomass as do events of extreme rainfall [22, 26].

Primary production (including that of MPB) is an Essential Biodiversity Variable (EBV), key to understand patterns and changes in the Earth's biodiversity [27, 28]. At longer time scales, biomass, and production of microphytobenthos can increase due to eutrophication [29], decrease due to increased turbidity caused by dredging activities [30] and increase or decrease due to increasing summer temperatures [31–33]. Furthermore, long-term changes in tidal amplitudes, wind stress and sea level affect sediment composition of tidal flats [34]. This implies that impacts of climate change, like temperature rise as well as the forecasted increase in the frequency of extreme events (heat waves, storm floods and downpours) and sea level

rise will affect the availability of microphytobenthos as a food source for higher trophic levels and thus the base for ecosystem services.

Monitoring of MPB is commonly performed in field campaigns, where samples are taken to determine surface chlorophyll-a concentrations as an index of biomass [35, 36] and for incubation with [14]C to determine production [37]. These measurements are, however, both logistically difficult and expensive to sustain as part of long-term monitoring programs and provide information valid for a very small spatial (less than 1 km$^2$) and temporal scale (a few times a year) only [38]. Alternatively, satellite data can be used to estimate phytobenthic biomass and productivity. Although with greater errors than the in-situ techniques [39], remote sensing provides synoptic monitoring of large areas at unparalleled spatial and temporal scales, even after considering the effect of cloud cover. Satellite-derived chlorophyll data, in combination with solar insolation, can be subsequently used to estimate the primary production by MPB [35, 36, 40].

Satellite-derived benthic chlorophyll-a concentration of the mudflats (mg m$^{-2}$) is generally based upon an empirical relationship with the Normalized Difference Vegetation Index (NDVI). NDVI is a widely used index in land remote sensing to monitor vegetation [41, 42]. This index is based upon the reflectance in a red and a near-infrared band. Driven by user needs, all Landsat and Sentinel 2 sensors include bands that allow NDVI derivation [43–45]. These band setups vary slightly across sensors and so there is not a single NDVI definition, which is relevant to multi-sensor monitoring. However, sensor specific NDVI's can be easily related to a common currency, like a reference sensor configuration or a derived quantity like chlorophyll concentration.

NDVI and MPB concentrations correlated well in some studies [23, 36, 46, 47], but this relationship was only moderate in others [48, 49] or varied between areas [35]. The relationship appears to be linear for most cases, with intercepts and slopes varying for different seasons and for the way in which NDVI and/or chlorophyll-a concentrations (CHLa) were determined. [49] found, however, an exponential relationship for the low range of values (NDVI<0.12, CHLa <80 mg m$^{-2}$) whilst others observed saturation of NDVI at higher (> 100 mg m$^{-2}$) chlorophyll-a concentrations [35, 46–48]. Part of this variation may be since chlorophyll-a field data are generally corrected for the presence of its degradation product pheophytin-a, while satellite-derived chlorophyll-a estimates are based on the sum of both pigments [50]. Variation in the contributions of chlorophyll-a and pheophytin-a to total pigment concentrations might interfere with the relation of chlorophyll-a and NDVI.

Satellite-derived production estimates of benthic algae (mg C m$^{-2}$ y$^{-1}$) are primarily based upon the benthic chlorophyll-a density (mg m$^{-2}$) of the mudflats. On intertidal mudflats, migrating benthic diatoms (epipelic) are the most important primary producers [51]. In the dark and when the flats are immersed, these diatoms are uniformly distributed over the top 2 mm of the sediment [51]. Depending on the species, algal cells appear at the surface between 0.5 and 2 h after exposure [52]. To estimate benthic primary production for a location, information on microphytobenthic biomass (mg CHLa m$^{-2}$) must be combined with an estimation of emersion time of the tidal area, available light conditions, and temperature [36]. To determine light conditions available for benthic production, an assumption must be made on the vertical distribution of ambient light and of the benthic algae in the top layer of the sediment [36, 51–53].

The aim of this study is to test generic methods to determine biomass and productivity of microphytobenthos by means of satellite-derived information in a highly turbid and heterotrophic intertidal ecosystem. Hereto, in-situ measurements of spectral reflectance (350–950 nm), of microphytobenthos (chlorophyll-a concentrations), of productivity estimates (by means of [14]C incubations) and of sediment characteristics (median grain size, mud content) were performed in the north-eastern part of the Netherlands during three seasons (autumn: September

2018, spring: April 2019 and summer: July 2019) over a range of sediment conditions. We aimed to:

- Explain the previously and presently observed variation in relationships between field-derived NDVI values and chlorophyll-a concentrations of microphytobenthos;

- Explore the potential of satellite (Sentinel 2) information to describe seasonal variation in spatial patterns of microphytobenthic biomass;

- Determine the most important sources of variation of outcomes of earth-observation based estimates of productivity of microphytobenthos;

- Describe potential long-term variation in biomass and productivity of MPB by comparing present results by previous findings.

## Material and methods

### Study area

The study area is the Dollard, the innermost part of the Ems estuary, which is enclosed between the Netherlands in the west and Germany in the east (Fig 1). The Dollard has a surface area of 103 km$^2$ of which 81% consists of intertidal flats, with sediments having a median grain size of 83 μm and a mud content of 41% [54]. The seasonal variation in the discharge of the river Ems (25 to 390 m$^3$ s$^{-1}$; [55]) results in salinities varying between freshwater to brackish (6–15 practical salinity units (psu)) and brackish to marine (15–25 psu) [54]). In 1975, the local production of microphytobenthos in the Dollard, with a peak in spring, added up to 9.3 10$^6$ kg C year$^{-1}$ [56]. Between 1976 and 1978, the annual primary production by MPB in the Dollard ranged, on average, from 71 to 232 gC m$^{-2}$, with highest values close to the mouth of the Westerwoldsche A [57]. In 2013, annual averaged biomass of microphytobenthos in the Dollard ranged between 10 and 75 mg chlorophyll a m$^{-2}$, with highest values (> 100 mg chlorophyll a m$^{-2}$) found during the spring bloom in the central part [58].

The Ems estuary has undergone large human-induced changes in geomorphology with consequences for the sediment dynamics. During the past decades to centuries, reduction of the intertidal area due to land reclamations and the deepening of estuarine tidal channels resulted in an increase in fine sediment import [59]. The subsequent increase in turbidity because of the loss of these sediment sinks [60] has caused concern and large-scale measures are being put into place including the construction of artificial saltmarshes at the expense of tidal flat systems [61].

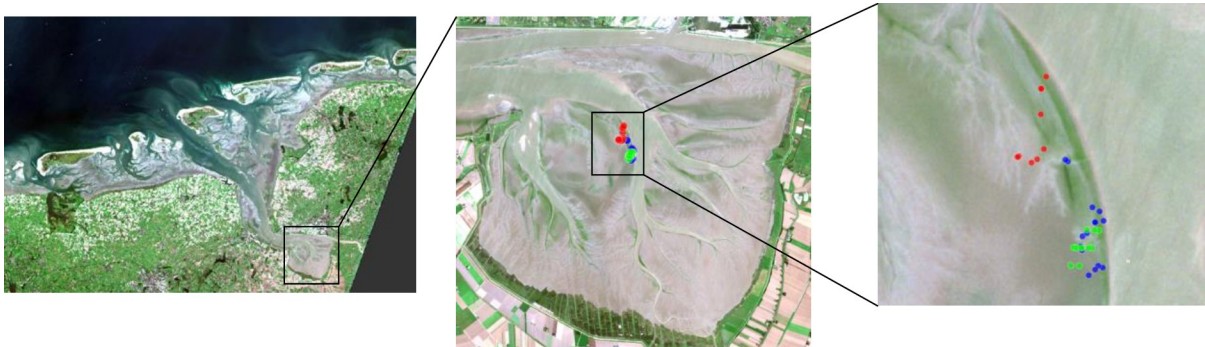

**Fig 1. Sampling stations.** Overview of sampling stations in the Ems estuary on 18 September 2018 (blue), 11 April 2019 (green) and 24 July 2019 (red), over a true colour image generated from a Level 2 Sentinel 2A image is taken on 18 September 2018 (Source: Copernicus).

Although it appears that the increased turbidity has resulted in a reduction of pelagic primary production [58], the historical and future effects on the benthic production remains unclear.

A permit for sampling and measuring in this part of the Wadden Sea was granted to the Royal NIOZ by the province of Friesland, the Netherlands. Sampling took place on the Heringsplaat in the Ems-Dollard estuary during low tide, on 18 September 2018, 11 April 2019, and 24 July 2019 on 9–15 stations per sampling date (Fig 1). The sampling area was near Station 5 of [56]. Each measurement was geo-positioned using a Trimble GEO XT GPS (0.5 m accuracy). Samples were collected to determine chlorophyll-a concentration as well as sediment characteristics. For each sampling period, [14]C-incubations were executed to establish microphytobenthic production at two stations.

## Sediments

Samples for grain size analysis were taken using a cut-off syringe (diameter 26 mm, area 530 mm$^2$), per station 3 cores of 2 mm thickness were collected and stored in a plastic container. Sediment samples were freeze-dried and homogenised. Grain sizes were analysed using a particle size analyser (Coulter LS 13 320) allowing the analysis of particles in the range 0.04–2000 μm divided into 126 size classes. The analysis followed a 'biological approach', meaning that organic matter and calcium carbonate were not removed [62]. Mud is defined as the fractions <63 μm (volume), and mud percentages were calculated as the contribution of this fraction to the total volume.

## Pigments

Samples for benthic chlorophyll-a were taken with a cut-off syringe (*see* Sediments), per station 3 cores of 2 mm thickness were collected and stored in a plastic container. Samples were transported in a cooling box (max. 2 hours) on mudflat and in -80˚C freezer during the car trip (max. 3 hours). At the laboratory each sample, consisting of three cores, was placed in a freezer (-80˚C) until analyses.

Before analysis, the samples were defrosted and 20 ml of 90% acetone was added to them, suspensions were mixed and stored overnight at 4˚C in the dark. The next day, samples were homogenized, and 8 ml of sample extract was centrifuged at 3000 RCF for 10 min. From the sample extract, 3 ml was pipetted in a cuvette and measured using a fluorescence spectrophotometer (F-2500 Hitachi). When concentrations were too high to be measured, samples were diluted with 90% acetone. Samples were measured again after adding 2 drops of 10% HCl.

Following [63], concentrations of uncorrected (not acidified) chlorophyll-a concentrations (CHLa_u; mg m$^{-2}$), corrected chlorophyll-a concentrations (CHLa_c; mg m$^{-2}$) and pheophytin-a (PHEOa; mg m$^{-2}$) were determined using the following equations:

$$CHLa\_u = Rb * F_s * \left( \frac{v_{extraction}}{v_{sample}} \right) * DF \tag{1}$$

$$CHLa\_c = (Rb - Ra) * \left( \frac{r}{r-1} \right) * F_s * \left( \frac{v_{extraction}}{v_{sample}} \right) * DF \tag{2}$$

$$PHEOa = (r * Ra - Rb) * \left( \frac{r}{r-1} \right) * F_s * \left( \frac{v_{extraction}}{v_{sample}} \right) * DF \tag{3}$$

In these equations, *Rb* (unitless) is the fluorescence signal of the sample before adding the acid solution; *Ra* (unitless) is the fluorescence signal after adding the acid solution, $V_{extraction}$ is the volume acetone added to the sample (ml), $V_{sample}$ is the sampled area (cm²) and *DF* is the dilution factor (unitless).

The other parameter values ($F_s$ and *r*) are derived from calibrations that were performed once for each sampling period. Hereto, fractions (20, 50, 100, 150, 200, 400 µl of standard (*Anacystis nidulans*; 3300 CHLa$_s$ µg l$^{-1}$) were added to 3 ml 90% acetone to create a dilution series (CHLa$_{sd}$). For each of the samples of this series, fluorometer readings were performed before (R$_{sdb}$) and after (R$_{sda}$) acidification. For each of these samples, the response factor $F_{sd}$ and the ratio r$_{sd}$ were respectively calculated as:

$$F_{sd} = CHLa_{sd} / R_{sdb} \qquad (4)$$

$$r_{sd} = R_{sdb} / R_{sda} \qquad (5)$$

Subsequently, $F_s$ and r were calculated as the average values for the dilution series.

## 14C uptake rate

Samples for primary production measurements were taken on two stations, that were also sampled for chlorophyll-a and sediment, by scraping off the top 1 mm of five sediment cores (3.14 cm² per core). Samples were taken to the lab (in the dark, in cool box). From each sample, approximately 2 ml (10% of core surface, two small spoons) was diluted in 75 ml Whatman© GF/F-filtered seawater from the same location. The sample was well mixed to produce a homogenous slurry. From this slurry, samples were taken for chlorophyll-a concentration and for the $^{14}$C uptake rate. All activities took place in dim light. The chlorophyll-a concentration of the slurry (µg l$^{-1}$) was determined as described above using triplicates using 10 ml of slurry instead of the sediment cores.

The incubation procedure followed that of [64], with small modifications. In the radioisotope laboratory, each slurry was well agitated while 2 ml subsamples were pipetted into 20 ml glass incubation vials. Per sample, 11 vials were filled, to each vial 50 µl of NaH$^{14}$CO$_3$ was added (approximate activity: 1.05 MBq ml$^{-1}$). To determine the actual activity added to the samples, controls (triplicates) were prepared for each sampling date by adding 50 µl of NaH$^{14}$CO$_3$ to 2 ml of NaOH. These samples were not incubated. All vials were placed in a (CHPT©, model TGC1000, equipped with 2 halogen light bulbs: Philips 13095, 250 W) photosynthetron [65], with one vial per sample being incubated in the dark, and all other vials placed at light intensities from 65 to 1522 µE m$^{-2}$ s$^{-2}$. Actual light (PAR) received in the incubators was measured inside the vials filled with 2 ml sample at each position using a light meter (WALZ ULM-500) with spherical micro sensor (US-SQS/L). There is a constant water flow around the vials in the photosynthetron, enabling a constant, set temperature during the incubation. Temperatures were set to in-situ water temperatures as measured during sampling. The carbon incorporation was stopped after 30 minutes by adding 100 µl of concentrated HCl (37%) to each vial except the controls, to remove all the non-incorporated inorganic carbon. The $^{14}$C method gives a good approximation of net production for most species [66].

The samples were counted using a scintillation counter (PerkinElmer, Tri-Carb 2910TR) including quenching correction, after the addition of 10 ml UltimaGold scintillation fluid to each vial. To check for possible light attenuation when counting ('quenching') due to the thick slurry, samples were counted again after diluting the slurry to see if this changed the count, which it did not.

## Daily primary production rate

The carbon fixation rate (P; mg C l$^{-1}$ h$^{-1}$) per sample was calculated according to the formula below [67]:

$$P = \left( \frac{(\mathrm{dpm_{sample}} - \mathrm{dpm_{dark}}) \times \mathrm{DIC} \times 1.05 \times \mathrm{T_{corr}} \times c}{\mathrm{dpm_{added}} \times t} \right) \tag{6}$$

For each sample, the corrected rate of disintegrations per minute (dpm) was calculated as the measured dpm of that sample ($dpm_{sample}$) minus the average dpm of the two dark flasks ($dpm_{dark}$). $DIC$ is the concentration of dissolved inorganic carbon (mg C l$^{-1}$).

To correct for a temperature difference between the in-situ temperature and the temperature during the incubation, a correction factor was applied:

$$\mathrm{T_{corr}} = e^{0.0693} \times (\mathrm{T_{in-situ}} - \mathrm{T_{incubation}}) \tag{7}$$

The constant 1.05 in Eq (2) is a factor to correct for the lower uptake rate of $^{14}$C compared to $^{12}$C, $c$ is a constant with value 1000 to convert units, $dpm_{added}$ is the dpm as measured in the control bottles corrected for the volume used and $t$ is the duration of the incubation (in hours). The fixation rates were normalised to chlorophyll-a concentrations (from the slurry, see previous paragraph) and with these rates P-E curves were fitted [68, 69]. P-E curves were fitted using a model described by [70].

To calculate the production of each sampling day, information on irradiance during the sampling day, the light attenuation in the sediments, the total amount of chlorophyll-a in the top 2 mm of sediment and the distribution of chlorophyll-a in the sediment is needed. It was assumed that production only took place when the sediments were emerged [18]. The height of the Heringsplaat (sampled tidal flat) is between +70 and -10 cm Amsterdam Ordnance Datum (NAP) [71]. Using water height data (RWS, station Nieuw Statenzijl), it was determined when the tidal flats were emerged during the sampling date, using a height of < 40 cm NAP (the average height of the Heringsplaat).

Downwelling PAR data during these emerged hours was recorded at a nearby station, named 'Nieuw Beerta', by the Royal Netherlands Meteorological Institute (www.knmi.nl), and provided as J m$^{-2}$ s$^{-1}$ (= W m$^{-2}$), which was converted in μE m$^{-2}$ s$^{-1}$ by multiplying this value by 4.66 and by 0.45 to obtain the fraction of solar light between 400 and 700 nm (PAR) [72]. PAR attenuation rate in the sediment (K$_d$; units mm$^{-1}$) was not measured (see Discussion) but estimated using an empirical relation between K$_d$ and chlorophyll-a concentrations of the sediment (Eq 4) described in [36].

$$\mathrm{K}_d = -3.1 + 1.8 \, x \ln CHLa_c \tag{8}$$

Note that Eq (4) uses the natural logarithm 'ln' instead of 'log' as is written in [36].

With respect to the vertical distribution of chlorophyll-a, it is generally assumed that most chlorophyll-a is confined to the top 2 mm of the sediment [51], but there is no agreement on how to model the vertical distribution of chlorophyll-a in this layer. Therefore, the following models of chlorophyll-a distribution were compared:

1. The chlorophyll-a concentration exponentially decreases with depth [51];

2. There is a uniform distribution of chlorophyll-a distribution over the top 2 mm of the sediment in the absence of light, when sediment is exposed, all chlorophyll-a from the top 1 mm migrates above and concentrates at the 0.2 mm layer [52];

3. The distribution of chlorophyll-a in the sediment depends on the mud content of the sediment [53], where we used the fraction (0–1) of mud is used and not the percentage (1–100%) as is mentioned in [53].

Daily production rates were estimated using the R package 'phytotools' [68, 69]. The maximum depth over which the production was integrated was set at 2 mm. Here, the chlorophyll-a concentration with depth a matrix 'cz' is used. This matrix should consist of two columns, the first with the chlorophyll-a concentration, the second with depth and not as it states in the description of 'phytotools' [69], the other way around.

## Normalized difference vegetation index (NDVI)

Before sediment samples were taken, the normalized difference vegetation index (NDVI) of the mudflats was determined by means of hyperspectral radiometers for each of the sampling stations. The radiometric measurements were performed using RAMSES radiometers (TriOS Mess- und Datentechnik GmbH). Downwelling irradiance ($E_s$; units W m$^{-2}$) and radiance ($L_u$; units W m$^{-2}$ sr$^{-1}$) sensors, respectively were installed on a portable frame and controlled with a field laptop. Spectral data were interpolated at 1 nm (Fig 2). The surface information, contained in $L_u$, was normalized by $E_s$ to derive the surface reflectance:

$$\rho = \frac{\pi L_u}{E_s}$$
(9)

Here, $\rho$ is unitless due to the scaling factor $\pi$ (sr), that accounts for the conversion between radiance and irradiance.

The normalized difference vegetation index (NDVI; unitless) was calculated according to:

$$NDVI = \frac{\rho(NIR) - \rho(R)}{\rho(NIR) + \rho(R)}$$
(10)

Here $\rho(NIR)$ and $\rho(R)$ refer to the reflectance values in the near-infrared and red, respectively [42]. The red and NIR wavelengths were set at 675 nm and 750 nm, respectively, following [35].

The NDVI definition is not uniform in the literature, even for hyperspectral data. For multispectral data, users are constrained to each sensor band setting. Because of this, there may be some concerns on the uniqueness of the CHL to NDVI relationships in the literature, which is affected by how NDVIs from different sensor configurations relate to each other. To clarify this matter, in-situ hyperspectral reflectance was resampled to the spectral bands of red and near-infrared of Landsat-7 Enhanced Thematic Mapper Plus (L7 ETM+), Landsat-8 Operational Land Imager (L8 OLI) and Sentinel-2 Multispectral Instrument (S2 MSI) by convoluting the reflectance with the corresponding relative spectral responses (RSRs) and NDVI was determined from them (Table 1 and Fig 2). CHLa (+PHEOa) was regressed to NDVI from each of these band settings to obtain sensor-specific algorithms.

## Spatiotemporal dynamics of MPB biomass in satellite images

The surface reflectance, from which NDVI is calculated, is found in atmospherically corrected data, namely Level 2. Level 2 images were available in the rolling archive at the Copernicus Open Access Hub, back until April 2018. For accessing older images, a request to the Long Term Archive is needed, but at the moment of writing this manuscript, such service was down. Alternatively, data were requested at the DIAS ONDA, processed until Level 1. These images were carefully processed to Level 2 consistently with the downloaded Level 2 products (http://step.esa.int/main/third-party-plugins-2/sen2cor/sen2cor_v2-8/). In particular, the

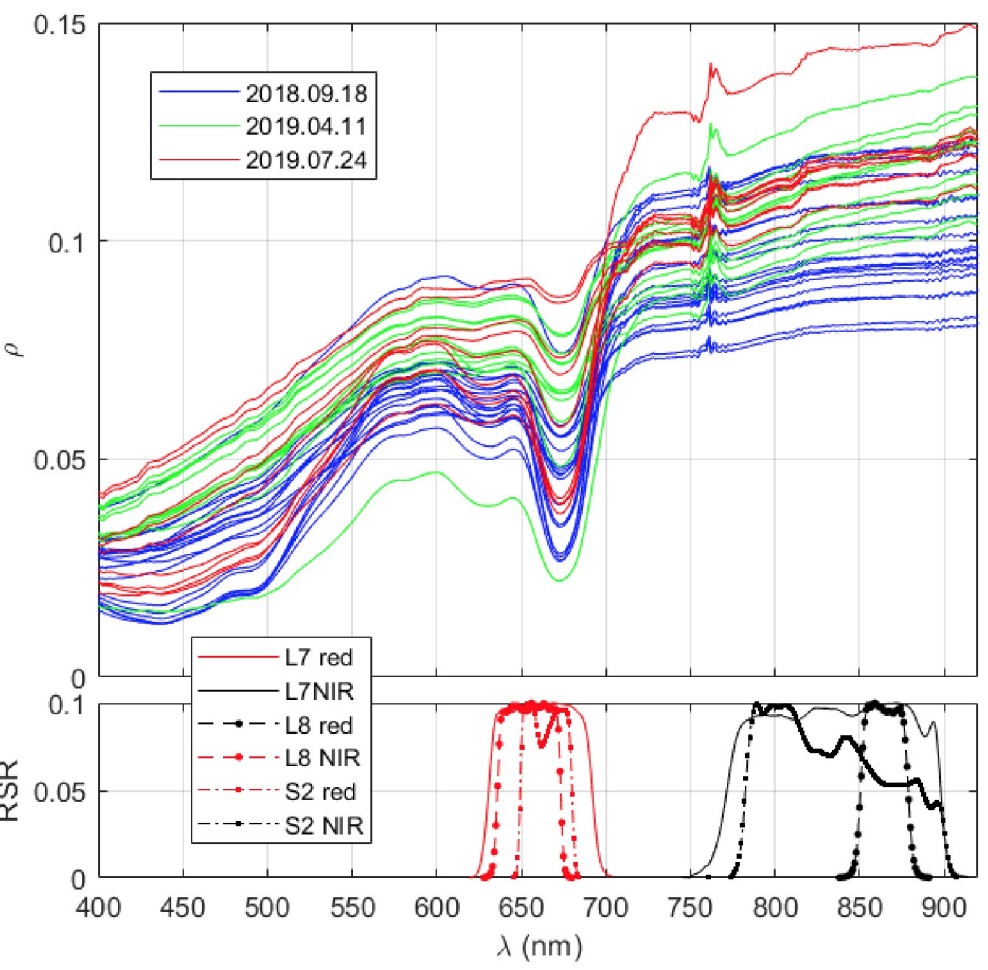

| Sensor | Red | | Near-infrared | |
|---|---|---|---|---|
| CW | | FWHM | CW | FWHM |
| L7 | 661 | 630-690 | 835 | 770-900 |
| L8 | 655 | 636-673 | 865 | 851-879 |
| S2 | 665 | 646-685 | 864 | 848-881 |

**Fig 2. Spectral reflectance and bands.** Spectral reflectance (top panel) of sediment as determined by means of hyperspectral sensors during the field surveys and spectral bands (bottom panel) in red and near-infrared (NIR) of the Landsat-7 Enhanced Thematic Mapper Plus ETM+ (L7), the Landsat 8 Operational Land Imager (L8) and the Sentinel 2 Multispectral Instrument (S2 File) based upon the centre wavelengths (CW; nm) and full width at half maximum (FWHM; nm) of these satellites.

Sen2cor v2.8 software was used with default options, making use of the ESA-CCI LC package to improve the accuracy of Sen2Cor classification over water, urban and bare areas and also to have a better handling of false detection of snow pixels (https://www.esa-landcover-cci.org/?q=node/189).

From the Level 2 images, the bands B4 (red) and B8 (NIR), relevant for the NDVI calculations, were selected and subset over the Dollard area. Yet, despite the atmospheric correction, in order to provide comparable data over all day times and seasons, the reflectance must be

**Table 1. Relationships between benthic pigment concentrations as a function of the NDVI reported in other studies.**

| Intertidal area | Sensor | Pigment | Method | Intercept | Slope | n | r² | p | Source |
|---|---|---|---|---|---|---|---|---|---|
| Bourgneuf Bay* (F) | GER 3700 | Chla_c | HPLC | 46 [b] | 444 [b] | NA | 0.97 | <0.001 | [46] |
| Bourgneuf Bay (F) | ASD | Chla_c | HPLC | 540 [b] | 1129 [b] | NA | 0.95 | <0.001 | [47] |
| Eastern Scheldt YB (NL) | RAMSES | CHLa_u | Spectro | 59 | 685 | 76 | 0.53 | <0.001 | [35] |
| Eden estuary EA (UK) | RAMSES | CHLa_u | Spectro | 55 | 555 | 74 | 0.68 | <0.001 | [35] |
| Eden estuary EB (UK) | RAMSES | CHLa_u | Spectro | 91 | 442 | 54 | 0.50 | <0.001 | [35] |
| E&W Scheldt (NL) | RAMSES | CHLa_u | Spectro | 30 | 556 | 138 | 0.75 | <0.001 | [36] |
| Saemangeum (KOR) | Landsat7 | CHLa_c | Spectro | 8 [b] | 229 [b] | 40 | 0.34 | <0.001 | [49] |
| Saemangeum (KOR) | Landsat7 | CHPH_c | Spectro | 2 [b] | 658 [b] | 40 | 0.59 | <0.001 | [49] |
| Sydney (AU) | ASD | CHLa_u | Spectro | 26 [a] | 252 [a] | 75 | 0.48 | <0.001 | [48] |
| Tagus estuary (P) | SPOT-HRV | CHLa_c | Spectro | -45 [b] | 390 [b] | 69 | 0.7 | <0.05 | [23] |
| Western Scheldt YA (NL) | MMS-1 | CHLa_u | Spectro | 21 | 437 | 77 | 0.66 | <0.001 | [35] |
| Western Scheldt YC (NL) | MMS-1 | CHLa_u | Spectro | 30 | 495 | 29 | 0.51 | <0.001 | [35] |

Linear relationships between benthic pigment concentrations (mg m$^{-2}$) as a function of the Normalized Difference Vegetation Index (NDVI) based upon 675nm (red) and 750nm (near-infrared) width using different hyperspectral (GER 3700 spectroradiometer, ASD FieldSpec 3FR, RAMSES-ARC-VIS), and satellite (Landsat7, SPOT) sensors with pigments of intertidal areas determined as uncorrected chlorophyll-a (CHLa_u), corrected chlorophyll-a (CHLa_c) and the sum of corrected chlorophyll-a and pheophytin-a (CHPH_c) concentrations (mg m$^{-2}$). Pigment analysis methods applied were high-performance liquid chromatography (HPLC) or spectrophotometry (Spectro).

* Cultivated after sampling;

[a] derived from a graph in the paper;

[b] intercept and slope recalculated based upon non-linear relationship in the original paper (NB: n, r² and p values are based upon the original relationship of NDVI with log-transformed pigment data).

corrected for bidirectional reflectance (BRDF) effects over non-Lambertian surfaces. For this purpose, the mean zenith and azimuth view angles and the zenith and azimuth sun angles were selected as well and resampled to the resolution of bands B4 and B8 (10 m). The BRDF correction followed the Ross-Thick-Li-Sparse reciprocal model, operationally applied to MODIS data [73], and tailored to Sentinel 2 data [74]. From BRDF-corrected surface reflectance, Eq (10) was applied to calculate NDVI, which can be transformed to surface pigment concentration, based on the field-derived relationships. Within the rectangular bounding box, these calculations were performed for pixels defined as mudflats, according to shapefiles provided by the Directorate-General for Public Works and Water Management (Rijkswaterstaat).

The updated bathymetry of the mudflats, provided by Rijkswaterstaat at 30 m resolution, was resampled to the satellite resolution and used to discriminate the satellite data based on it. Here, we defined two intertidal zones, namely "high" mudflats as those 40 cm or more above NAP and "low" mudflats being those below NAP +40 cm.

## Results

### Environmental conditions

**Sediments.** Considering all sampling periods, the median grain size (MGS) at Heringsplaat varied between 19 μm and 150 μm (Table 2), which would classify as 'silt' and 'fine sand' respectively [75]. Between sampling periods, the median grain size at the sampled stations was relatively low (88 ± 41 μm; n = 15) in September 2018, intermediate (94 ± 31 μm; n = 12) in July 2019 and relatively high (118 ± 10 μm; n = 9) in April 2019 (Table 2). The coefficient of variation (CV) of the MGS was highest (46%) in September 2018, intermediate (33%) in July 2019 and lowest (9%) in April 2019 (Table 2).

Considering all sampling periods, the fraction of mud (%mud) varied between 8% and 75% (Table 2). Between sampling periods, the %mud at the sampled stations was relatively high in July 2019 (39 ± 16%) and September 2018 (37 ± 23%) and relatively low (18 ± 8%) in April 2019 (Table 2). The coefficient of variation of the %mud was highest (62%) in September 2018 and relatively low in April 2019 (46%) and July 2019 (41%) (Table 2).

For each of the sampling periods, the median grain size (MGS) and fraction mud of the sediment were highly negatively correlated (-0.98 ≤ r ≤ -0.91; Fig 3).

**Corrected chlorophyll-a.** Considering all sampling periods, the corrected chlorophyll-a concentrations (CHLa_c) at Heringsplaat varied between 55 mg m$^{-2}$ and 399 mg m$^{-2}$ (Table 3). Between sampling periods, CHLa_c at the sampled stations was relatively high (183 ± 100 mg m$^{-2}$; n = 15) in September 2018 and comparably low in April 2019 (107 ± 34 mg m$^{-2}$; n = 12) and July 2019 (103 ± 47 mg m$^{-2}$; n = 9) (Table 3). The coefficient of variation of CHLa_c was highest (54%) in September 2018, intermediate (46%) in July 2019 and lowest (32%) in April 2019 (Table 3).

In September 2018, corrected chlorophyll-a concentrations (mg CHLa_c m$^{-2}$) at Heringsplaat were positively correlated with median grain size (r = 0.68; n = 15) and negatively correlated with mud content of the sediment (r ≤ -0.75) (Fig 3). No significant correlations between corrected chlorophyll-a concentrations and these sediment characteristics were found for April and July 2019 (Fig 3).

**Pheophytin-a.** Considering all sampling periods, the pheophytin-a concentrations (PHEOa) at Heringsplaat varied between 8 mg m$^{-2}$ and 99 mg m$^{-2}$ (Table 3). Between sampling periods, PHEOa at the sampled stations was relatively high (63 ± 27 mg m$^{-2}$; n = 15) in September 2018, intermediate in April 2019 (42 ± 19 mg m$^{-2}$; n = 9) and lowest in July 2019 (29 ± 9 mg m$^{-2}$; n = 9) (Table 3). The coefficient of variation of PHEOa was comparably high in September 2018 (43%) and July 2019 (45%) and lowest (31%) in April 2019 (Table 3).

**Table 2. Sediment characteristics.**

| Station | Sep-18 MGS | %mud | Lat. | Lon. | Apr-19 MGS | %mud | Lat. | Lon. | Jul-19 MGS | %mud | Lat. | Lon. |
|---|---|---|---|---|---|---|---|---|---|---|---|---|
| 1 | 81 | 44 | 53.29.394 | 7.16.015 | 112 | 27 | 53.17.738 | 7.09.581 | 128 | 13 | 53.18.141 | 7.09.243 |
| 2 | 71 | 48 | 53.29.403 | 7.15.972 | 114 | 19 | 53.17.731 | 7.09.561 | 87 | 38 | 53.18.087 | 7.09.231 |
| 3 | 125 | 14 | 53.29.381 | 7.15.946 | 117 | 13 | 53.17.725 | 7.09.544 | 100 | 32 | 53.18.024 | 7.09.266 |
| 4 | 129 | 8 | 53.29.349 | 7.15.888 | 116 | 19 | 53.17.728 | 7.09.528 | 95 | 41 | 53.17.978 | 7.09.198 |
| 5 | 129 | 9 | 53.29.488 | 7.15.819 | 120 | 20 | 53.17.727 | 7.09.490 | 116 | 27 | 53.17.991 | 7.09.229 |
| 6 | 123 | 13 | 53.29.508 | 7.15.807 | 119 | 19 | 53.17.727 | 7.09.452 | 103 | 34 | 53.18.000 | 7.09.126 |
| 7 | 84 | 45 | 53.29.563 | 7.15.819 | 124 | 11 | 53.17.691 | 7.09.442 | 77 | 53 | 53.17.997 | 7.09.119 |
| 8 | 69 | 49 | 53.29.582 | 7.15.860 | 126 | 11 | 53.17.655 | 7.09.431 | 120 | 40 | 53.18.227 | 7.09.239 |
| 9 | 19 | 75 | 53.29.643 | 7.15.934 | 126 | 10 | 53.17.647 | 7.09.478 | 23 | 70 | 53.18.267 | 7.09.265 |
| 10 | 48 | 54 | 53.29.653 | 7.16.010 | 125 | 12 | 53.17.709 | 7.09.462 | | | | |
| 11 | 61 | 50 | 53.29.653 | 7.16.011 | 89 | 39 | 53.17.706 | 7.09.485 | | | | |
| 12 | 27 | 71 | 53.29.705 | 7.15.972 | 124 | 15 | 53.17.734 | 7.09.519 | | | | |
| 13 | 79 | 48 | 53.29.727 | 7.15.906 | | | | | | | | |
| 14 | 150 | 8 | 53.29.976 | 7.15.667 | | | | | | | | |
| 15 | 128 | 22 | 53.29.987 | 7.15.643 | | | | | | | | |
| AVG | 88 | 37 | | | 118 | 18 | | | 94 | 39 | | |
| SD | 41 | 23 | | | 10 | 8 | | | 31 | 16 | | |
| CV | 46% | 62% | | | 9% | 46% | | | 33% | 41% | | |

Median grain size in (MGS; μm), mud fraction (<63 μm) of the sediment (%mud) and the GPS- coordinates for the sampling locations at the three field campaigns. Shaded cells indicate stations for which carbon-fixation rates were measured as well.

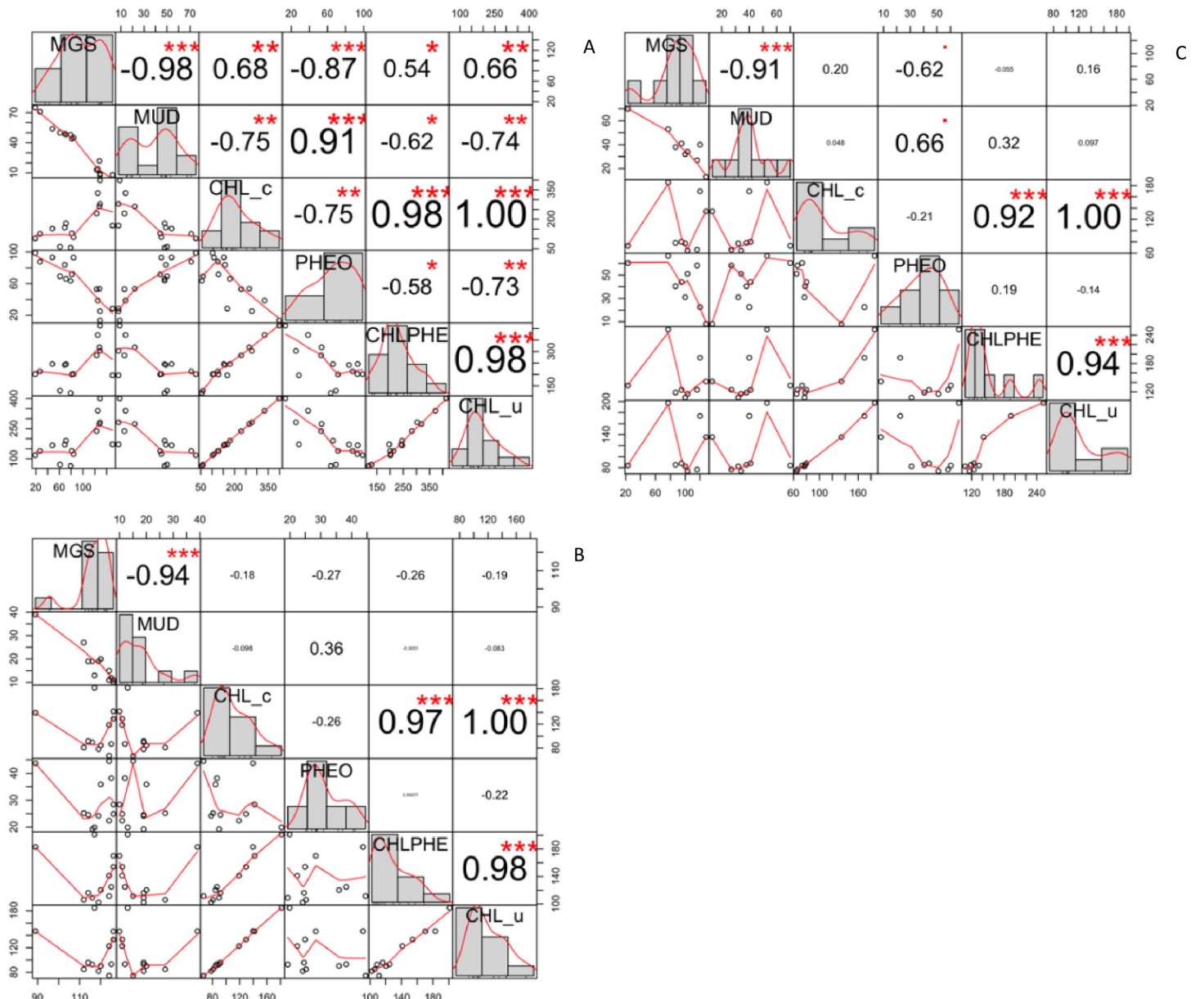

**Fig 3. Correlations between algal pigments and sediment characteristics.** Correlations between pigment concentrations (mg m$^{-2}$), median grain size (MGS; μm) and mud content of the sediment (MUD; %) at the mudflats in the Ems estuary in September 2018 (n = 15) (top panel), April 2019 (n = 12) (middle panel) and July 2019 (n = 9) (bottom panel). Pigments comprise corrected chlorophyll-a (CHLa_c), corrected pheophytin-a (PHEOa_c), the sum of corrected chlorophyll-a and pheophytin (CHLPH_c), and uncorrected chlorophyll-a (CHLa_u).

In September 2018, pheophytin-a concentrations (mg PHEOa m$^{-2}$) at Heringsplaat were negatively correlated with median grain size (r = -0.87; n = 15) and highly positively correlated with mud content of the sediment (r $\leq$ 0.91) (Fig 3). No significant correlations between pheophytin-a concentrations and these sediment characteristics were found for April 2019, but correlations were relatively high (-0.62 with MGS and 0.66 with mud content) for July 2019 (Fig 3).

**Sum of corrected chlorophyll-a and pheophytin-a.** Considering all sampling periods, the sum of corrected chlorophyll-a and pheophytin-a (CHLPH_c) concentrations at Heringsplaat varied between 102 mg m$^{-2}$ and 412 mg m$^{-2}$ (Table 3). Between sampling periods, CHLPH_c at

**Table 3. Chlorophyll pigments.**

| | Sept 2018 | | | | April 2019 | | | | July 2019 | | | |
|---|---|---|---|---|---|---|---|---|---|---|---|---|
| Station | CHLa_c | PHEOa_c | CHLPH_c | CHLa_u | CHLa_c | PHEOa | CHLPH_c | CHLa_u | CHLa_c | PHEOa_c | CHLPH_c | CHLa_u |
| 1 | 128 | 72 | 200 | 140 | 81 | 25 | 106 | 85 | 134 | 8 | 142 | 136 |
| 2 | 179 | 66 | 245 | 190 | 92 | 27 | 119 | 96 | 78 | 41 | 119 | 85 |
| 3 | 334 | 39 | 373 | 340 | 181 | 20 | 201 | 185 | 77 | 31 | 108 | 83 |
| 4 | 278 | 25 | 303 | 281 | 90 | 19 | 109 | 93 | 80 | 44 | 124 | 88 |
| 5 | 399 | 14 | 413 | 400 | 84 | 36 | 120 | 91 | 66 | 58 | 124 | 77 |
| 6 | 230 | 53 | 283 | 238 | 78 | 24 | 102 | 82 | 65 | 51 | 116 | 74 |
| 7 | 108 | 93 | 201 | 123 | 119 | 22 | 141 | 123 | 186 | 66 | 252 | 198 |
| 8 | 158 | 84 | 242 | 171 | 129 | 25 | 154 | 133 | 170 | 23 | 193 | 174 |
| 9 | 102 | 99 | 201 | 118 | 142 | 28 | 170 | 147 | 73 | 60 | 133 | 84 |
| 10 | 152 | 93 | 245 | 167 | 87 | 38 | 125 | 93 | | | | |
| 11 | 60 | 69 | 129 | 71 | 139 | 44 | 183 | 147 | | | | |
| 12 | 126 | 88 | 214 | 140 | 67 | 45 | 112 | 75 | | | | |
| 13 | 55 | 64 | 119 | 65 | | | | | | | | |
| 14 | 168 | 29 | 197 | 172 | | | | | | | | |
| 15 | 265 | 55 | 320 | 273 | | | | | | | | |
| AVG | 183 | 63 | 245 | 193 | 107 | 29 | 137 | 112 | 103 | 43 | 146 | 111 |
| SD | 100 | 27 | 82 | 96 | 34 | 9 | 33 | 34 | 47 | 19 | 47 | 47 |
| CV | 54% | 43% | 33% | 50% | 32% | 31% | 24% | 30% | 46% | 45% | 32% | 42% |

Concentrations of corrected benthic chlorophyll-a (CHLa_c; mg m$^{-2}$), corrected benthic pheophytin-a (PHEOa_c; mg m$^{-2}$), the sum of corrected benthic chlorophyll-a and benthic pheophytin-a (CHLPH_c; mg m$^{-2}$) and uncorrected benthic chlorophyll-a (CHLa_u; mg m$^{-2}$) for the sampling locations at the three field campaigns. Shaded cells indicate stations for which primary production was measured as well. GPS-coordinates of each station at each sampling date can be found in the caption of Table 2.

the sampled stations was relatively high in September 2018 (245 ± 82 mg m$^{-2}$; n = 15) and comparably low in April 2019 (137 ± 33 mg m$^{-2}$; n = 12) and relatively low in July 2019 (146 ± 47 mg m$^{-2}$; n = 9) (Table 3). The coefficient of variation of CHLPH_c was comparably high in in September 2018 (33%) and July 2019 (32%), and lowest (24%) in April 2019 (Table 3).

The average fraction of pheophytin-a compared to the average sum of corrected chlorophyll-a and pheophytin-a was relatively high in July 2019 (29%), intermediate in September 2018 (26%) and relatively low in April 2019 (21%) (Table 3).

For all three sampling periods, the sum of corrected chlorophyll-a and pheophytin-a (mg CHLPH_c m$^{-2}$) at Heringsplaat was positively correlated with corrected chlorophyll-a concentrations (Fig 3 and S1 File). Only for September 2018, also a positive relationship of this sum with pheophytin-a was found (Fig 3).

**Uncorrected chlorophyll-a.** For all three sampling periods, the uncorrected chlorophyll-a concentrations (CHLa_u) were highly correlated with those of corrected chlorophyll-a (Fig 3), with the corrected chlorophyll-a concentrations (CHLa_c) being 9.03 ± 1.39 mg m$^{-2}$ (Sept 2018), 6.19 ± 1.65 mg m$^{-2}$ (April 2019) and 8.73 ± 3.27 mg m$^{-2}$ (July 2019) lower than the uncorrected chlorophyll-a concentrations (Table 3).

## Production

Between sampling periods, water and air temperatures were relatively high (20 and 36.3˚C respectively) in July 2019, intermediate in September 2018 (16.6 and 22.8˚C) and relatively low (7.6 and 9.6˚C) in April 2019 with in July a difference of 16.3˚C between water temperature

and maximum air temperature (Table 4). Concentrations of dissolved inorganic carbon were comparable between sampling periods, ranging between 28.3 mg l$^{-1}$ in July 2019 and 29.3 mg l$^{-1}$ in September 2018 (Table 4). The attenuation coefficient $K_d$ in the sediment as derived from benthic chlorophyll-a concentrations (following [36]) ranged between 4.7 m$^{-1}$ in July 2019 to 6.4 m$^{-1}$ in September 2018 (Table 4).

The light-production curves that were fitted for each of the three sampling dates and two stations showed that the slope of the light-limited part of the curve ($\alpha^\beta$; mg C (mg CHLa_c)$^{-1}$ h$^{-1}$ (PAR µE m$^{-2}$ s$^{-1}$)$^{-1}$) and the maximum photosynthetic production rate ($P^\beta_{max}$; mg C (mg CHLa_c)$^{-1}$ h$^{-1}$) were highest at Station 8 in July and lowest at Station 11 in April (Fig 4 and Table 4). The values of $\alpha^\beta$ and $P^\beta_{max}$ were positively correlated with each other ($r^2 = 0.99$, n = 6, S2 File).

The daily benthic production rates (mg C m$^{-2}$ d$^{-1}$) were highest ($> 339.6$ mg C m$^{-2}$ d$^{-1}$) on 24 July 2019 in station 8 and lowest ($< 1$ mg C m$^{-2}$ d$^{-1}$) on 11 April 2019 for station 11 (Table 5). Daily production rates varied for different assumptions with respect to vertical distribution of benthic chlorophyll-a in the sediment, with model 1 always supplying the highest values followed by model 2 (88% ± 10% compared to model 1) and model 3 (62% ± 18% compared to model 1) (Table 5). The largest relative difference between models was found on 18 September for station 14, with the values for model 3 being 58% lower than those for model 1 (Table 5).

No significant relationship was found between daily production rates (mg C m$^{-2}$ d$^{-1}$) and corrected chlorophyll-a concentrations (mg m$^{-2}$) for any of the vertical distribution models, with the variation in biomass of benthic algae explaining 23% (model 3) to 27% (model 2) of the variation in production rates (S3 File).

## NDVI to CHLa (+PHEOa) relationships

Considering all sampling periods, the normalized difference vegetation index (NDVI) as determined by the hyperspectral sensors (based upon reflectance at 675 and 750 nm; Fig 2) varied between 0.093 and 0.596 (Table 6 and S4 File). Between sampling periods, the NDVI at the sampled stations was relatively high in September 2018 (0.338 ± 0.143), intermediate in July 2019 (0.224 ± 0.141) and relatively low (0.294 ± 0.171) in April 2019 (Table 6). The coefficient of variation of the NDVI was highest in April 2019 (63%), intermediate in July 2019 (58%) and relatively low (42%) in September 2018 (Table 6).

For April 2019 and July 2019, respectively, a significant ($p < 0.01$) and an almost significant ($p < 0.1$) relationship between chlorophyll-a concentrations (mg CHLa m$^{-2}$) with the normalised difference vegetation index (NDVI) was found (Table 7). The values of the intercepts

**Table 4. Overview of water quality parameters and photosynthetic parameters.**

| Sampling date | Station | WT (˚C) | AT (˚C) | DIC (mg l$^{-1}$) | $K_d$ (mm$^{-1}$) | $\alpha^\beta$ | $P^\beta_{max}$ | $E_k$ |
|---|---|---|---|---|---|---|---|---|
| 18 Sept. 18 | 12 | 16.6 | 22.8 | 29.3 | 5.8 | 0.010 ± 0.001 | 4.43 ± 0.23 | 441 ± 0 |
| 18 Sept. 18 | 14 | 16.6 | 22.8 | 29.3 | 6.4 | 0.007 ± 0.003 | 1.44 ± 0.34 | 197 ± 1 |
| 11-Apr-19 | 11 | 7.6 | 9.6 | 29.7 | 5.9 | 0.0002 | 0.05 | 229 ± 1 |
| 11-Apr-19 | 12 | 7.6 | 9.6 | 29.7 | 4.6 | 0.011 ± 0.003 | 1.19 ± 0.16 | 108 ± 0 |
| 24-Jul-19 | 8 | 20 | 36.3 | 28.7 | 6.2 | 0.052 ± 0.005 | 21.71 ± 1.21 | 421 ± 0 |
| 24-Jul-19 | 9 | 20 | 36.3 | 28.3 | 4.7 | 0.012 ± 0.004 | 2.48 ± 0.57 | 211 ± 0 |

Water temperature (WT), air temperature (daily max) (AT) derived from a local weather station (Nieuw Beerta; www.knmi.nl), dissolved inorganic carbon (DIC) and attenuation coefficient in the sediment ($K_d$) for the sampling dates and stations for which primary production of microphytobenthos was calculated. The EP-model gives as output $\alpha^\beta$ which is the slope of the light-limited part of the curve (mg C (mg CHLa_c)$^{-1}$ h$^{-1}$ (PAR µE m$^{-2}$ s$^{-1}$)$^{-1}$) and $P^\beta_{max}$ is the maximum photosynthetic production rate in (mg C (mg CHLa_c)$^{-1}$ h$^{-1}$). The minimum saturation light intensity $E_k$ is derived as $P_{max} / \alpha$ (PAR µE m$^{-2}$ s$^{-1}$) [76].

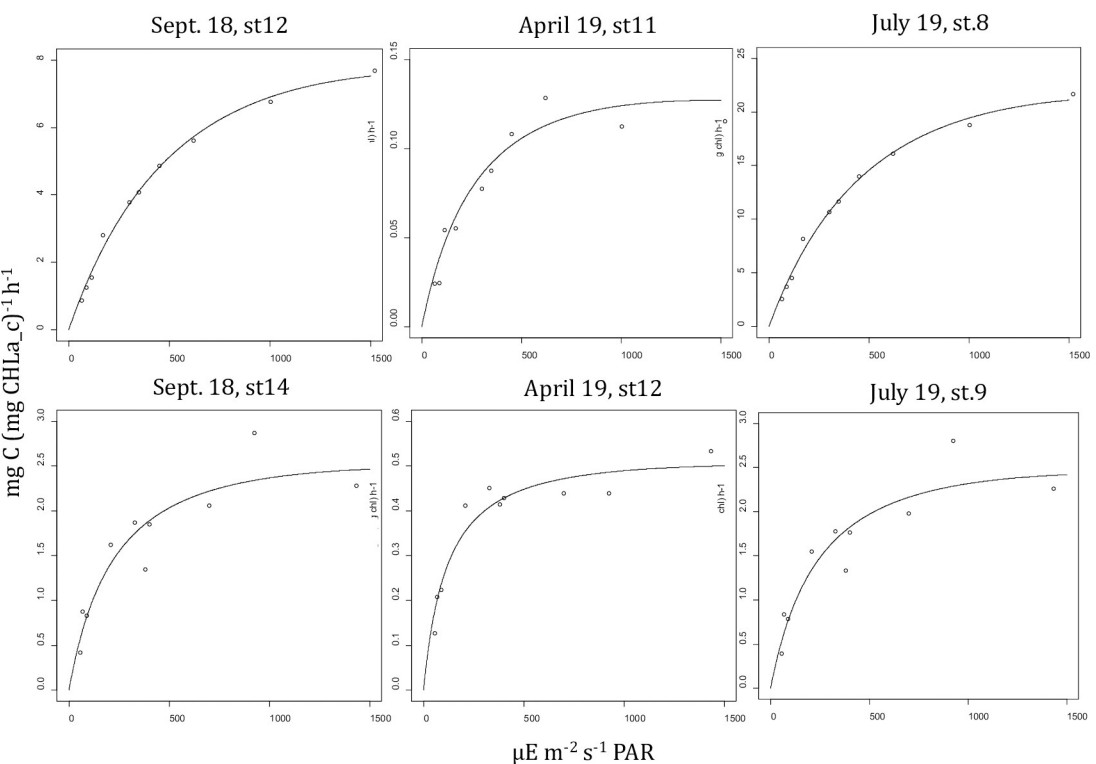

**Fig 4. Carbon fixation rates.** The calculated production rate (mg C (mg CHLa_c)$^{-1}$) h$^{-1}$) of microphytobenthos at the different light intensities (dots) and the modelled light-production curves (lines). The model fit was made using the EP- model [70]. Note the different scales used for the different dates and stations.

(65 ± 13; 54 ± 27) and slopes (118 ± 49; 166 ± 81) of these relationships were comparable for these two sampling periods (Table 7).

For September 2018, no such relationship was found. For this reason, to determine the NDVI to CHLa relationship that will be applied to satellite data only data collected in April and July 2019 will be used. Relationships of corrected benthic chlorophyll-a concentrations as a function of NDVI and showed that the values of NDVI as determined by means of hyper-spectral sensors (based upon reflectance at 675 and 750 nm) were higher than the NDVIs as based upon red and near-infrared (NIR) spectral bands of the Landsat 7 ETM, the Landsat 8 OLCI and the Sentinel 2 (Table 7).

**Table 5. Daily carbon fixation rates.**

| Sampling date | Station Nr | Station Code | Emersion time (h) | Daily carbon fixation rate mg C m$^{-2}$ d$^{-1}$ | | |
|---|---|---|---|---|---|---|
| | | | | Model 1 | Model 2 | Model 3 |
| 18 Sept. 18 | 12 | A | 9 | 61.6 | 58.4 | 55.1 |
| 18 Sept. 18 | 14 | B | 9 | 41.7 | 38.2 | 17.4 |
| 11 April 19 | 11 | A | 8 | 0.65 | 0.57 | 0.43 |
| 11 April 19 | 12 | B | 8 | 13.3 | 9.7 | 7.0 |
| 24 July 19 | 8 | A | 8 | 516.5 | 487.6 | 339.6 |
| 24 July 19 | 9 | B | 8 | 43.4 | 33.2 | 38.4 |

Daily carbon fixation rates (mg C m$^{-2}$ d$^{-1}$) on sampling dates. The time production could have taken place (emersion time) is also given in hours. The vertical distribution of chlorophyll-a in the sediment was estimated using three different models (see Material and methods).

**Table 6. Normalized difference vegetation indices.**

| Station | Sept 2018 | | | | April 2019 | | | | July 2019 | | | |
|---|---|---|---|---|---|---|---|---|---|---|---|---|
| | hss | L7 | L8 | S2 | hss | L7 | L8 | S2 | hss | L7 | L8 | S2 |
| 1 | 0.398 | 0.306 | 0.305 | 0.349 | 0.125 | 0.165 | 0.182 | 0.175 | 0.267 | 0.240 | 0.249 | 0.268 |
| 2 | 0.213 | 0.183 | 0.184 | 0.204 | 0.274 | 0.256 | 0.266 | 0.282 | 0.093 | 0.131 | 0.144 | 0.137 |
| 3 | 0.208 | 0.179 | 0.180 | 0.200 | 0.596 | 0.501 | 0.506 | 0.554 | 0.096 | 0.134 | 0.148 | 0.140 |
| 4 | 0.222 | 0.189 | 0.190 | 0.212 | 0.165 | 0.183 | 0.197 | 0.198 | 0.176 | 0.190 | 0.202 | 0.205 |
| 5 | 0.183 | 0.164 | 0.167 | 0.182 | 0.140 | 0.172 | 0.188 | 0.183 | 0.164 | 0.186 | 0.199 | 0.199 |
| 6 | 0.323 | 0.263 | 0.264 | 0.295 | 0.171 | 0.194 | 0.209 | 0.208 | 0.408 | 0.342 | 0.348 | 0.384 |
| 7 | 0.368 | 0.289 | 0.287 | 0.327 | 0.165 | 0.188 | 0.202 | 0.201 | 0.530 | 0.430 | 0.429 | 0.480 |
| 8 | 0.595 | 0.427 | 0.417 | 0.493 | 0.163 | 0.191 | 0.207 | 0.204 | 0.472 | 0.368 | 0.370 | 0.420 |
| 9 | 0.443 | 0.321 | 0.316 | 0.372 | 0.180 | 0.200 | 0.214 | 0.215 | 0.441 | 0.355 | 0.359 | 0.402 |
| 10 | 0.558 | 0.398 | 0.390 | 0.462 | 0.151 | 0.179 | 0.194 | 0.191 | | | | |
| 11 | 0.209 | 0.182 | 0.186 | 0.204 | 0.410 | 0.352 | 0.358 | 0.391 | | | | |
| 12 | 0.480 | 0.347 | 0.342 | 0.402 | 0.146 | 0.173 | 0.187 | 0.185 | | | | |
| 13 | 0.213 | 0.176 | 0.178 | 0.200 | | | | | | | | |
| 14 | 0.196 | 0.174 | 0.176 | 0.192 | | | | | | | | |
| 15 | 0.460 | 0.360 | 0.357 | 0.405 | | | | | | | | |
| AVG | **0.338** | **0.264** | **0.263** | **0.300** | **0.224** | **0.230** | **0.242** | **0.249** | **0.294** | **0.264** | **0.272** | **0.293** |
| SD | **0.143** | **0.092** | **0.088** | **0.109** | **0.141** | **0.100** | **0.096** | **0.113** | **0.171** | **0.111** | **0.106** | **0.130** |
| CV | **42%** | **35%** | **34%** | **36%** | **63%** | **44%** | **40%** | **46%** | **58%** | **42%** | **39%** | **45%** |

NDVIs for the sampling locations at the three field campaigns, based upon spectral information from hyperspectral sensors using spectral bands for the hyperspectral sensors (hss), Landsat 7 (L7), Landsat 8 (L8) and Sentinel 2 (S2). Shaded cells indicate stations for which also primary production was measured.

For all three satellites under consideration, the relationships of the sum of corrected chlorophyll-a and pheophytin-a concentrations as a function of NDVI explained more of the variance (56%-57%) than those of relationships between corrected chlorophyll-a (44%) and uncorrected chlorophyll-a (47%) concentrations as a function of NDVI (Table 7). The results

**Table 7. Linear relationships between benthic pigment concentrations as a function of NDVI for this study.**

| Period | Pigments | Intercept | Slope | n | $r^2$ | p |
|---|---|---|---|---|---|---|
| Sept18 | CHLa_u | 251 ± 66 | -172 ± 180 | 15 | 0.07 | 0.358 |
| April19 | CHLa_u | 71 ± 13 | **186 ± 48** | 12 | 0.59 | 0.003 |
| July19 | CHLa_u | 60 ± 27 | 174 ± 79 | 9 | 0.41 | 0.063 |
| Sept18 | CHLa_c | 248 ± 68 | -193 ± 185 | 15 | 0.08 | 0.315 |
| April19 | CHLa_c | 65 ± 13 | **187 ± 49** | 12 | 0.59 | 0.004 |
| July19 | CHLa_c | 54 ± 28 | 169 ± 82 | 9 | 0.37 | 0.080 |
| Sept18 | PHEOa_c | 17 ± 13 | **134 ± 36** | 15 | 0.51 | 0.003 |
| April19 | PHEOa_c | 31 ± 5 | -7 ± 20 | 12 | 0.01 | 0.742 |
| July19 | PHEOa_c | 34 ± 14 | 30 ± 41 | 9 | 0.07 | 0.479 |
| Sept18 | CHLPH_c | 266 ± 57 | -60 ± 157 | 15 | 0.01 | 0.710 |
| April19 | CHLPH_c | 96 ± 12 | **180 ± 48** | 12 | 0.59 | 0.004 |
| July19 | CHLPH_c | 87 ± 24 | **199 ± 71** | 9 | 0.53 | 0.027 |

Linear relationships between benthic pigment concentrations (mg m$^{-2}$) as a function of the Normalized Difference Vegetation Index (NDVI) based upon 675nm (red) and 750nm (near-infrared) with benthic pigments as uncorrected chlorophyll-a (CHLa_u), uncorrected chlorophyll-a (CHLa_c), pheophytin (PHEOa_c) and the sum of corrected chlorophyll-a and pheophytin-a (CHLPH_c) concentrations (mg m$^{-2}$) of microphytobenthos at the mudflats in the Ems estuary. Significant relationships (without correction for multiple comparisons) are printed in bold.

are consistent with the fact that NDVI is more sensitive to CHLa and related decomposition pigments than to CHLa alone. For this reason, the sum of corrected chlorophyll-a and pheophytin-a concentrations were chosen as the reference data for the NDVI calibration and posterior application to satellite images.

## Spatiotemporal dynamics of MPB biomass in satellite images

Between January 2018 and February 2020, thirteen cloud-free Level-2 images and two Level-1 images during low tide were found (S5 File). Most of the images were taken around (n = 6) or shortly after (1 to 2 hours) the occurrence of the astronomical low tide (S6 File). Only the image of 21$^{st}$ April 2018 was taken one hour before astronomical low tide (S6 File). After processing the Level 1 images to Level 2 and BRDF adjustment, the relationship between the sum of benthic chlorophyll-a and pheophytin-a concentrations and NDVI as described in this paper was applied (Table 7).

Highest overall biomass was observed in spring (April/May) 2018 with average values of more than 150 mg (CHLa+PHEOa) m$^{-2}$, whilst lowest average biomass of less than 110 mg (CHLa+PHEOa) m$^{-2}$ occurred in early summer (7 June 2018, 25 June 2019) and late summer (18 September 2018, 26 August 2018) (Figs 5 and 6). Average concentrations were relatively high (more than 120 mg (CHLa+PHEOa) m$^{-2}$) in late winter (27 February 2019) (Figs 5 and 6). From this time series, it appears that the highest peak on 21 April (2018) was followed by a rapid decline in average concentrations of more than 100 mg (CHLa+PHEOa) m$^{-2}$ in less than 50 days (Fig 6).

Concentrations of microphytobenthos pigments were, on average, relatively high at the lower parts (below NAP +40 cm) of the tidal flats, except for 8 and 15 May 2018, 6 August 2018 and 18 September 2018 where highest average concentrations were found at the higher parts (> NAP +40 cm) of the Dollard tidal flats (Figs 5 and 6).

## Discussion

### Biomass

**Relationships between NDVI and pigment concentrations.** No significant relationship between chlorophyll-a concentrations (mg m$^{-2}$) and NDVI was found for the campaign in September 2018, in contrast to that found for April and July 2019 (Table 7). This could have been caused by a significant time lag (10–25 min) between the pigment sampling and NDVI measurements in September 2018. [13] performed a controlled experiment in which artificial light illuminating a microphytobenthos layer was switched on an off while NDVI was continuously monitored. A sharp increase in biomass after switching on the light was followed by a steady NDVI increase and vice versa when the lights were switched off. The phenomenon at the basis of this finding is the vertical migration of microphytobenthos and accumulation of cells at the surface. Indeed, during the few hours of low tide, the variability can be so large that a few minutes between the measurements of NDVI (directly at the sediment surface) and those of CHLa (summed for the top layer of the sediment of 2 mm) might be too much for a direct comparison. A high temporal variability in NDVI was also reported in [77]. Thus, during the field campaign in September 2018, vertical migration of diatoms during the time lag between sampling and NDVI measurements might have caused that the conditions during the CHLa and NDVI measurements were not the same anymore. In view of these negative results, the measurement design was changed for the subsequent campaigns for April and July 2019 where sampling of NDVI and pigments was done within one minute after each other. Considering April and July 2019, the relationship between chlorophyll-a concentrations (mg m$^{-2}$) and NDVI is remarkably constant with a slope ranging between 180 and 199 mg m$^{-2}$ per NDVI interval, depending

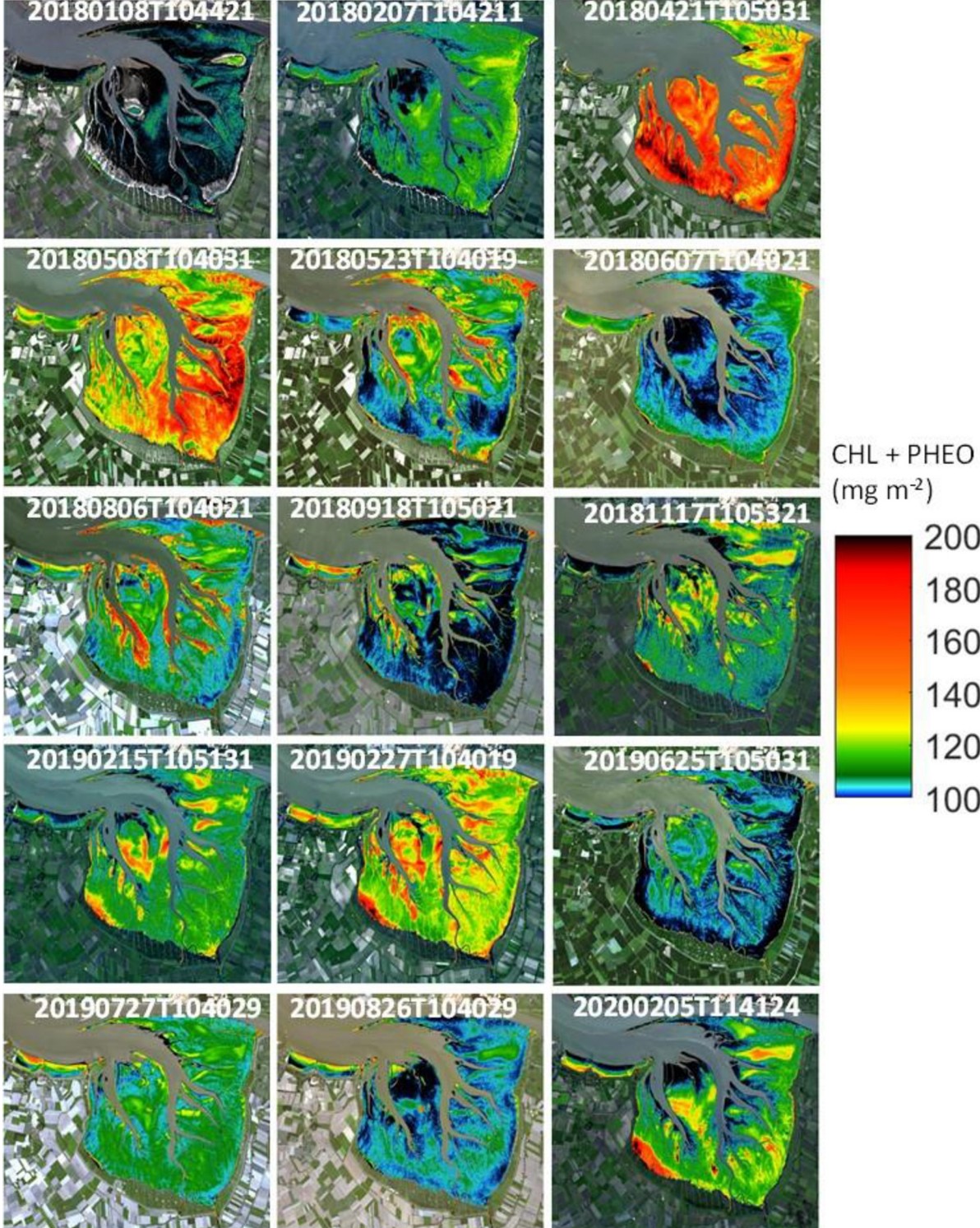

**Fig 5. Temporal and spatial distribution chlorophyll-a at the research area.** Concentrations of the sum of chlorophyll-a and pheophytin-a (mg m$^{-2}$) in the Dollard, the inner part of the Ems estuary, based upon a relationship with NDVI as derived from field measurements in April and July 2019 with spectral bands of Sentinel2 (CHLa+PHEOa = 242.63NDVI_s2 + 75.581; r$^2$ = 0.56). Derived from various Sentinel 2A and 2B images at various dates (see S5 and S6 Files). Images were generated from Level 2 Sentinel 2A images (Source: Copernicus).

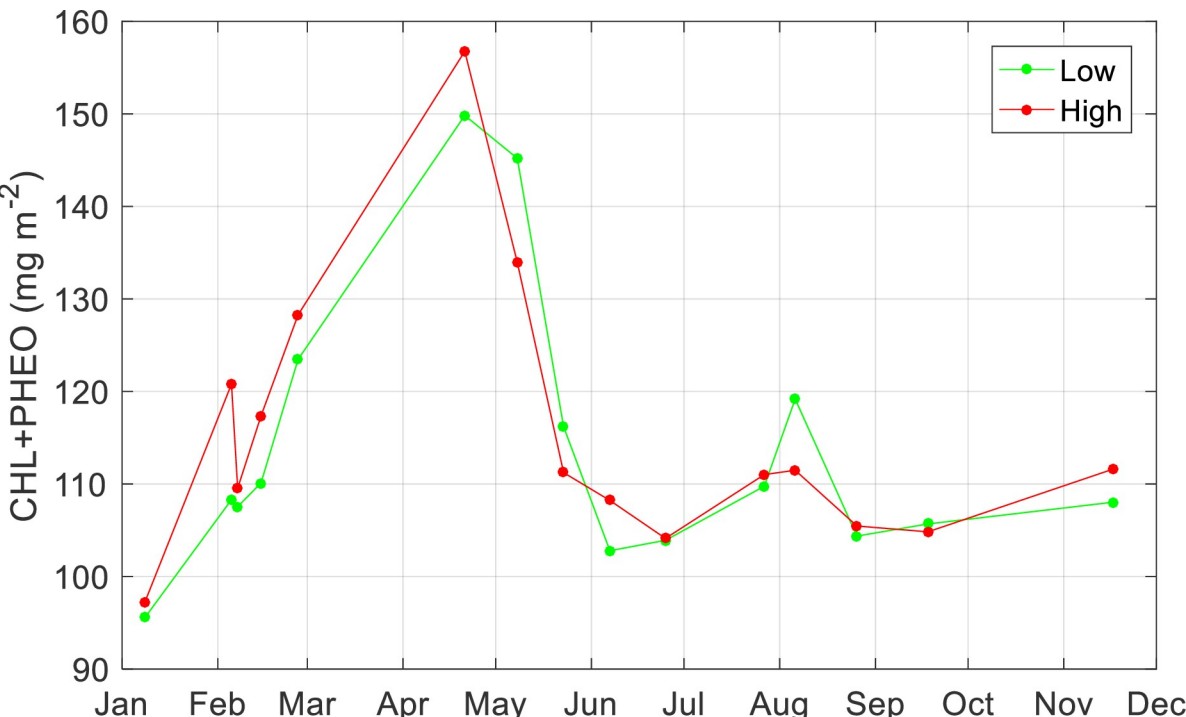

**Fig 6. Monthly variation in chlorophyll-a at the research area.** Average concentrations of the sum of chlorophyll-a and pheophytin-a (mg m$^{-2}$) in the Dollard, the inner part of the Ems estuary, based upon a relationship with NDVI as derived from field measurements in April and July 2019 with spectral bands of Sentinel2 (CHLa+PHEOa = 242.63NDVI_s2 + 75.581; r$^2$ = 0.56), for the tidal flat area lower than NAP+40 cm (green line) and for the tidal flat area higher than NAP+40 cm (red line). Derived from various Sentinel 2A and 2B Level 2 images at various dates (see Fig 5 and S5 File).

on the period and the pigments considered (Table 7). The values of these slopes are relatively low compared to the range (from 229 to 1129 mg m$^{-2}$) and average values (523 ± 236 mg m$^{-2}$) as found in other studies (Table 1). The total biomass in the sediment might have included subsurface cells that did not contribute to NDVI.

The highest correlation between NDVI and CHLa was obtained when CHLa was corrected (CHLa_c) and PHEOa was added (CHLPH). The results are consistent with the fact that NDVI is more sensitive to CHLa and related decomposition pigments than to CHLa alone [50], and these results highlights the contribution of other pigments in the NDVI signal. In the case that only monitoring of the live fraction of microphytobenthos is desired, relationships to only CHLa can be established as well, at the expenses of losing some degree of predictability. More attention should be given to develop algorithms that accurately estimate chlorophyll-a rather than the sum of chlorophyll-like pigments including phaeophytin and non-microphyto-benthos pigments like chlorophyll-b. The different band settings of all the multispectral sensors considered in this paper supposed significant differences in the slope and intercept of the NDVI to CHLa relationship, thus highlighting the need of specifying the band setting when the relationship is built. In the view of long-term multi-sensor monitoring, these particularities need to be considered in order to not introduce artificial biases in a time series. Care should also be taken when applying relations between NDVI and chlorophyll-a in other areas then the study area [78].

**Seasonal dynamics of NDVI.** Using the relation between chlorophyll-a and pheophytin-a with NDVI, 15 satellite images were processed (S7 File), showing the estimated concentration of both pigments for the Dollard area. These pictures clearly show relatively high concentrations

in winter months (November 2018, February 2019 & 2020), reaching peak concentrations in April and May (2018), while summer biomasses were relatively low (Figs 5 and 6).

In December 2018 and 2019, daily irradiances at noon fell below $E_k$ (248 ± 164 µE PAR m$^{-2}$ s$^{-1}$; Table 6) so light conditions were most probably too low to sustain net MPB growth (Fig 7). In February 2018, ice masses covered the tidal flats which may have limited light penetration, although MPB biomass (this paper) and growth has been observed under such circumstances [79]. In February 2019, when no ice was present, MPB biomass at the higher tidal flats was higher than the year before (Fig 6). Biomass of MPB peaked in April, which is in line by findings by [19]. From March to September, daily irradiances are always above the minimum required ($E_k$), but migratory species might retreat to deeper sediment layers to prevent photo-damages. This mechanism can result in lower production [80]. In January, February, October and November, growth would be stimulated during relative sunny days (Fig 7).

The relatively low MPB biomass found in summer months was already observed for the western Wadden Sea in the early 1970s [84]. [23] reported summer values of MPB in the Tagus estuary (Portugal), to be much lower than in winter which they explained being the result of thermo-inhibition of MPB photosynthesis due to extreme air temperatures during their study year (up to 5°C higher than the decadal summer average). Surface temperatures of 25°C and above are considered to inhibit growth of microphytobenthos [81, 82]. In 2018 and 2019 maximum air temperatures were above this threshold in the summer months for 39 and 36 days, respectively, implying that production on these days could have been reduced, likely explaining the observed summer dip (Figs 7 and 8). As sediment temperatures can reach even higher temperatures than air, the number of days with reduced production thus seem a cautious estimate.

Additionally, grazing may reduce biomass of microphytobenthos in particular during summer when grazing pressure is relatively high [85]. Potentially, grazing by macrofauna could be important [85], but looking at the biomass data of macrofauna in the Dollard (between 1.5 and 4.5 g AFDW m-2 for 2009–2019; [54] and NIOZ unpublished data) this impact may not be that relevant here. Grazing by meiofauna in the Dollard was high in the beginning of the 1980s, but at that time considered not high enough to impact on MPB production [86]. If meiofauna grazing pressure is still similar and if primary productivity has ceased (as suggested by [54] and [58]), however, grazing by meiofauna might be now a relevant driver for seasonality in MPB biomass.

Wind stress is generally higher during winter than during summer (Fig 7). Assuming a linear relationship between resuspension of benthic microalgae and wind speed [87] above a wind speed of 3 m s$^{-1}$ [83], then erosion driven by wind stress would take mainly place between October and March. It should be noted, however, the area is a very protected enclosed bay (Fig 1), and in addition, resuspension of MPB is modified by the growing phase of the MPB and the cumulative effects by other marine organisms that enhance of reduce erosion [88].

The number of rainy days (defined as > 10 mm rainfall in 24 hours; www.knmi.nl) was 13 in 2018 and 19 in 2019 (Fig 7). For the period 2018–2019, highest daily sum of rainfall occurred on 13 May 2018, with a total of 38.5 mm rain of which 18.4 mm fell between 17:00 and 18:00 local time (www.knmi.nl). At that day, astronomical low tide was at 18:16 local time, implying that the cloud burst most likely occurred when the tidal flats were fully exposed and subsequently most vulnerable to this extreme event. This freshening of the mudflats might have enhanced the strong decline in microphytobenthos as has been observed between 8 and 23 May 2018 (Figs 7 and 8).

The consistent seasonal pattern as has been observed for the Dollard in this study underlines the feasibility of using satellite images with a frequency of 5 to 10 images per year to monitor seasonal and, subsequently, year-to-year variation in biomass of microphytobenthos on

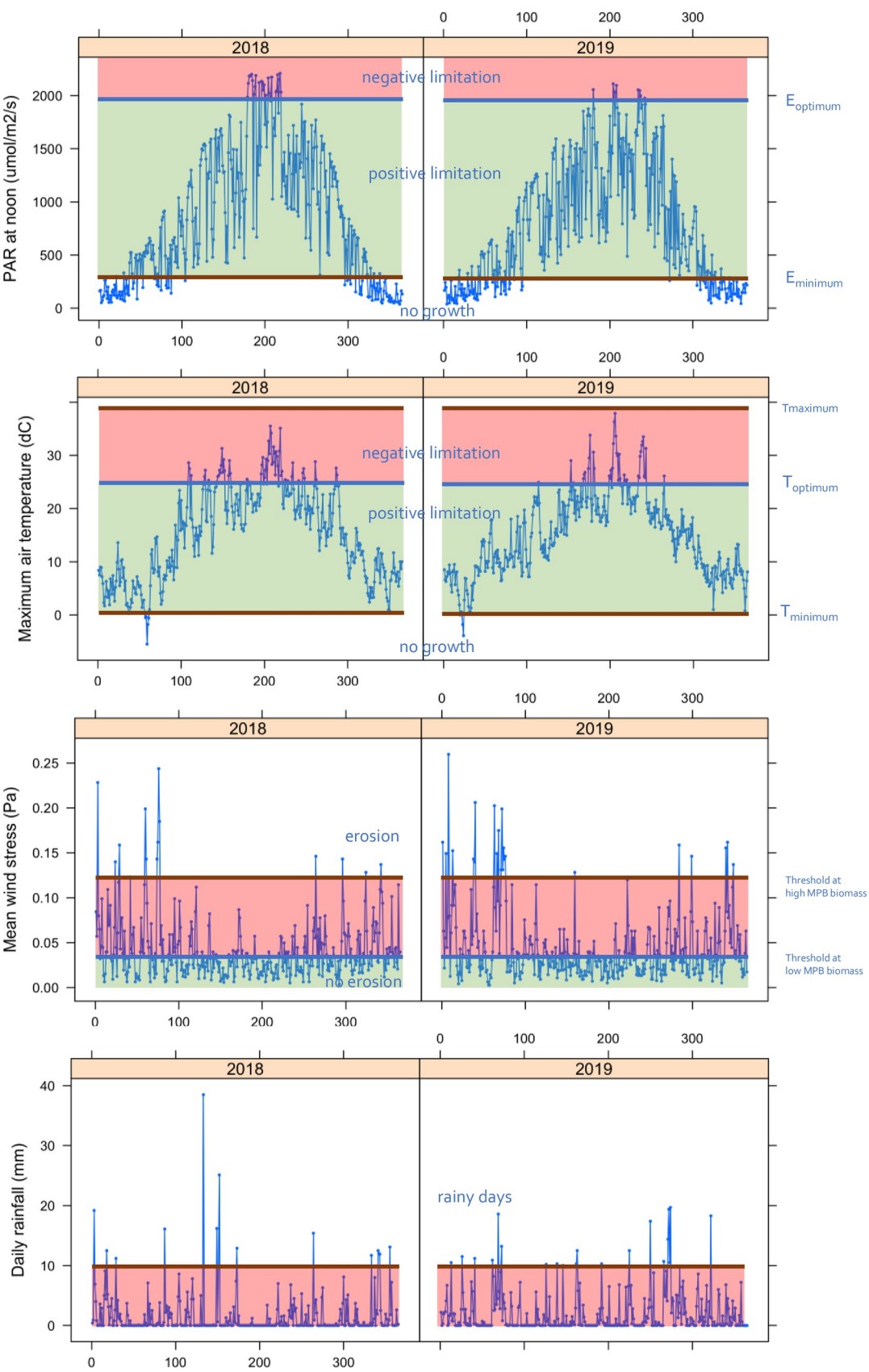

**Fig 7. Environmental conditions at the research area.** Environmental conditions in the Dollard as derived from a local weather station (Nieuw Beerta; www.knmi.nl). A) Insolation at solar noon (µE PAR m$^{-2}$ s$^{-1}$), with the horizontal lines indicating when light conditions are too low ($< E_k$; see Table 4) for MPB growth or higher ($> 2000$ µE PAR m$^{-2}$ s$^{-1}$ [80]) than required for optimal MPB growth. B) Daily maximum air temperature (˚C), with a horizontal line indicating when temperatures are too high for growth (thermo-inhibition) [81, 82]. C) Daily average wind speed data (m s$^{-1}$), with the horizontal line indicating the wind speed from where wind stress might result in resuspension of microbenthic algal mats [83]. D) Daily sum of rainfall (mm per day), where horizontal line indicates rainy days (rainfall $> 10$ mm per day; www.knmi.nl).

tidal flats of temperate coastal ecosystems. In particular if a long-term time series is being developed that enables decomposition of long-term trends and seasonal dynamics.

**Spatial patterns in NDVI.** In the current study at Heringsplaat, highest concentrations of microphytobenthos were generally found at the higher tidal flats (Figs 5 and 6), which is in line with findings by others [15, 19, 23, 55, 89]. The higher tidal flats are emerged for a longer period during low tide, resulting in a positive feedback between higher stability (low resuspension rate) of the sediment and a high biomass of benthic microalgae [19, 90].

The lack of a positive relationship between chlorophyll-a and the mud content of the tidal flats is, however, in contrast to findings in many other studies [16, 18–20]. Only [91] also reported on a positive relation between mud and pheophytin-a. Accumulation of fine sediments occurs often in 'depositional environments' where there is sedimentation of algae that

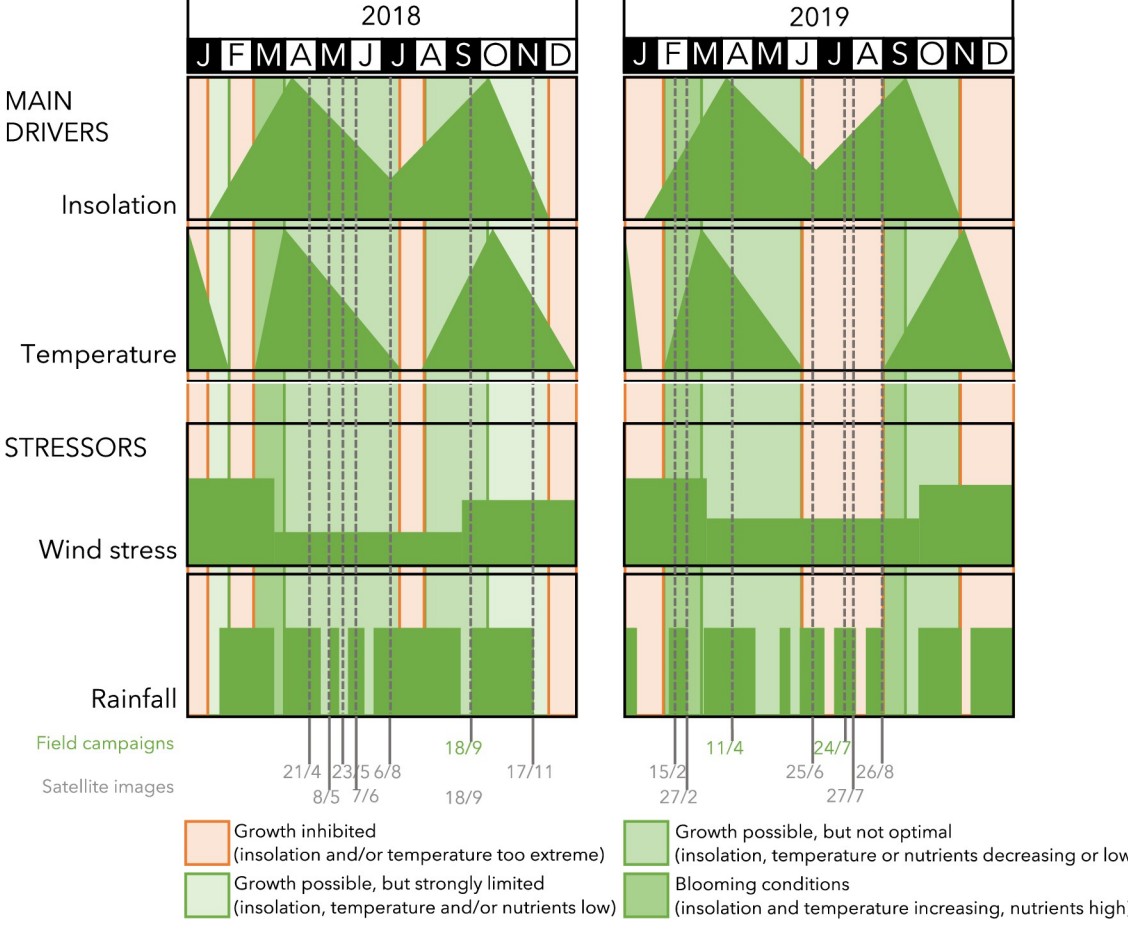

**Fig 8. Conceptual model.** Conceptual model of the main drivers and additional stressors of seasonality in local microphytobenthos biomass in shallow, silty, temperate tidal flat ecosystems.

resuspended from other habitats. Allochthonous detritus including (dead) phytoplankton cells accumulate in the Dollard [56], resulting is a high concentration of degradation pigments (pheophytin-a). Part of the NDVI measured in the Dollard area most likely did not originate from the MPB but from fresh organic matter imported from the more outward parts of the Ems estuary and possibly from the rivers. To resolve the contribution of the different pigments and their origin to the pool of chlorophyll-like components, HPLC analysis should be considered.

## Productivity

**Methodology.** In the current study benthic primary production rates were derived from [14]C uptake rates measured on resuspended sediments. There are drawbacks to this method, one of these is that the measurements are performed under artificial conditions and that the integrity of the sediment is disturbed. [92] compared a variable fluorescence technique with radio-isotope measurements on optically thin suspensions and concluded that both methods yielded comparable estimates of the photosynthetic parameters, but that there were significant differences between photosynthetic parameters measured on undisturbed sediment and in suspensions. The authors concluded that the application of variable fluorescence techniques in-situ on undisturbed sediments is limited due to complex effects of migration in undisturbed sediments and they suggested the method of measuring carbon fixation rates on slurries as the preferred method. Others [e.g., 78, 93] performed in-situ $CO_2$ exchange measurements and found a good correlation between variable fluorescence and production rates, but that relation depended on the location; For sites with a high MPB biomass the direct measurements of primary production were lower than was expected based on the variable fluorescence measurements. The authors hypothesised that part of the biomass is not taking part in the production, as micro-migration to the surface is prevented. In these situations, primary production estimates based on variable fluorescence would likely result in an overestimation and direct measurements of primary production are advised.

**Photoinhibition.** The EP-model allows for an estimate of a photo-inhibition parameter, but in the current study, photo-inhibition did not occur (Fig 4). The estimated photosynthetic parameters $\alpha^\beta$, $P^\beta_{max}$ and $E_k$ were very variable between stations and between dates, but they were within the range of values reported in previous studies [12, 64, 94]. The low values for $\alpha^\beta$ and especially $P^\beta_{max}$ found in April (Table 4) might have been the result of low temperatures as will be discussed below.

**Light attenuation.** Light attenuation in the sediment is notoriously difficult to measure in the field. A generally applied solution to come to an estimate of the light attenuation in the sediment is to suspend and resettle the upper mm of sediment over a light sensor [12, 64]. Reported attenuation coefficients obtained in this manner were between 3.5 and 7.9 mm$^{-1}$ [64, 95]. [96] measured attenuation coefficients using an optic fibre in cores made up of inorganic sediment that differed in grain size. The attenuation coefficients varied between 0.99 mm$^{-1}$ for sediments with a grain size between 250–500 μm and 3.46 mm$^{-1}$ for sediments with a grainsize <63 μm. Only occasionally, measurements have been performed on intact natural cores [36], with this method more realistic estimates of in-situ light attenuations are expected.

[36] measured attenuation and related this to several environmental variables including chlorophyll-a concentration and mud content. Chlorophyll-a concentration alone best described the measured attenuation coefficients. Reported attenuation coefficients here were roughly between 2.5 and 10.5 mm$^{-1}$. In the current study, this relation between chlorophyll-a concentration and light attenuation was used (c.f. [36]). Values estimated in the current study ranged from 4.6 to 6.4 mm$^{-1}$ (Table 4). This means that 90% of the light was lost within the first 500 μm of the sediment.

The modelled distribution in this layer will thus determine the potential production rate in each sediment as it influences how the biomass of microphytobenthos is distributed of the photic zone. In the current study, three different models were applied to describe the vertical distribution of chlorophyll-a in the sediment. Model 1, where the vertical distribution exponentially declined, always had the highest concentration in the first 500 μm and thus the highest daily production estimates. Model 3, which used the mud content of the sediment to describe the vertical distribution of chlorophyll-a in the sediment, always gave the lowest estimate of daily production (Table 5). In model 3, more chlorophyll-a accumulated in the top layer of the sediment, in case of a muddy sediment.

Previous studies reported a positive relation between the chlorophyll-a concentration in the first 2 mm of sediment and the mud content (e.g., [36, 53]). In those cases, sediments with high chlorophyll concentrations will occur at muddy sites, and most chlorophyll-a is expected, under the assumptions of model 3, in the first μm of the sediment. With an attenuation coefficient estimated based on chlorophyll-a concentration, light attenuation is estimated to be high, but production under these conditions can still be relatively high.

In the current study, however, the correlation between mud content and chlorophyll-a concentration was poor to negative (Fig 3). Using model 3, under conditions with a high chlorophyll-a concentration and a low mud content, a low mud content should result in a more homogenous distribution of chlorophyll-a in the sediment, but with a high attenuation coefficient due to the high chlorophyll-a concentration. On these occasions, the daily production is most likely to be under-estimated.

**Temperature.** The photosynthetic parameter $P^{\beta}_{max}$ was very low in on the sampling day in April (Table 4), resulting in a lower daily production rate (Table 5) then would be expected based on the chlorophyll-a concentration in de sediment (Table 3). A possible explanation for this low value of $P^{\beta}_{max}$ might be the minimum temperature of below zero at 10cm above the surface. This cold spell occurred after a mild period, with air temperature being >20˚C (www.knmi.nl). It is known that low temperatures have an impact on $P^{\beta}_{max}$ [82].

Temperature also plays a role in the measured rates in July. In the current study, samples were incubated to estimate uptake rates of $^{14}$C using in-situ water temperatures. Production rates were then calculated using these water temperatures. The sediment temperature of a tidal flat can differ substantially from water temperatures when emerged during daytime due to solar heat [81]. In the current study, sediment temperatures were not measured, but air temperatures give a better idea of temperatures experienced by the microphytobenthos than water temperatures. While microphytobenthos was incubated at 20˚C, the temperature experienced by the community was likely to be much higher with an air temperature of 36.3˚C (Table 4). Higher temperatures are expected to increase the photosynthetic rate until a maximum temperature is reached, at temperatures above the optimum temperature, rates will start to decrease [81]. [81] estimated the inhibitory temperature for microphytobenthos to be around 25˚C and recorded that sediment temperature could increase with a rate of 3˚C h$^{-1}$. In July, the in-situ water temperature was 20˚C and a decrease in production rate is expected during low tide due to suboptimal sediment temperatures and the daily rates might have been over-estimated.

## Monitoring

Matched optical chlorophyll and NDVI data from our field surveys showed that a relationship between both indexes is feasible, although more campaigns would be desirable to increase the robustness and confidence. The fact that the slope of the regression (Table 7) was lower than found by others (Table 1) might be explained by part of the standing stock of the microphytobenthos being deeper than 2 mm (the sampling depth). For the Dollard, the proportion of

microphytobenthos in the upper 5 mm ranged between 40% and 61% of total biomass in the 20 mm depth layer [89] underlying the need for more knowledge on depth profiles of microphytobenthos.

With the collected data set, estimating the daily production rate using the chlorophyll-a concentration resulted in 27% explained variance. To obtain a more reliable estimate of daily production rates, light attenuation coefficients and temperature need more attention. Regarding light attenuation, measurements should be performed using intact natural cores for many stations and the attenuation coefficients need to be related to sediment characteristics (c.f. [36]). It is recommended to additionally measure temperature in the sediment top layer. Modelling the vertical distribution by either exponential decrease or assume migration to the top 200 μm (c.f. [51, 52]) seems to be a reasonable assumption as the model of [53], which modelled the distribution of chlorophyll-a dependent of sediment type, gave unexpected results.

Since NDVI can also be estimated from satellite data, our results imply that it should be possible to use satellite images to estimate benthic primary production of relatively small and turbid estuaries. The Landsat sensors monitor at a spatial resolution of 30m and have been providing data since the mid 1980's at bi-weekly periodicity. Today, the European Sentinel-2 sensors as well as the ongoing Landsat missions 7 and 8, as well as the forthcoming Landsat 9, provide a very valuable dataset that has yet to be analysed and interpreted.

## Long-term variation

Based upon the three sampling campaigns, the average biomass of microphytobenthos was $131 \pm 45$ mg CHLa m$^{-2}$ (spectrophotometric; Table 3). Based upon the satellite images, annually averaged microphytobenthic biomass in the Dollard was approximately 115 mg CHLa +PHEOa m$^{-2}$ (Figs 5 and 6), and 86 mg CHLa m$^{-2}$ under the assumption that 75% of these pigments was CHLa (see Table 3).

In 1976–1978, average annual values for three sampling stations in the Dollard were $95 \pm 65$ mg CHLa m$^{-2}$ (spectrophotometric; [89]). In 1992–1999, the annual averages for these three stations were $98 \pm 56$ mg CHLa m$^{-2}$ (HPLC converted to spectrophotometric values; [31]). For the 1990s, a positive correlation was found between annual chlorophyll-a of microphytobenthos and annually averaged air temperatures [31]. Recalculation based upon the three Dollard stations for microphytobenthos (CHL$_{Dollard}$; [31]) and the Eelde weather station for air temperatures (AT$_{Eelde}$; cdn.knmi.nl), resulted in the following positive significant relationship: CHL$_{Dollard}$ = 20,2 AT$_{Eelde}$− 88,7 (n = 10, r$^2$ = 0,43, p = 0,039). Using the average air temperatures at Eelde in 2018 and 2019 (10,6˚C in both years; cdn.knmi.nl), the average annual CHL$_{Dollard}$ in 2018 and 2019 should have been 115 mg CHLa m$^{-2}$ which is higher than was measured for these years. Although it cannot be excluded that these differences in biomass are due to differences in sampling and analyses techniques, it could also imply that this relationship is no longer true, for example due to summer temperatures on the mudflats being higher than the optimum temperature for photosynthesis of 25˚C [31, 81]. With heat waves predicted to occur more frequently in the future [97, 98], this lack of fit for the relation between temperature and chlorophyll-a need further attention.

Based upon the three sampling campaigns, the average primary production of microphytobenthos ranged between 76 mg C m$^{-2}$ d$^{-1}$ and 113 mg C m$^{-2}$ d$^{-1}$, depending on the assumption on the vertical distribution of chlorophyll-a in the sediment (Table 5). If assumed that these values represent averages for the growing season and that the growing season runs from mid-January to mid-November (10 months), then the annual production of microphytobenthos in the Dollard would have been between 22 g C m$^{-2}$ y$^{-1}$ and 33 g C m$^{-2}$ y$^{-1}$. Another way of estimating the annual production was based on the relation with chlorophyll-a [57] for the same

area using the annual average chlorophyll-a concentration based on the satellite date (86 mg m$^{-2}$, corrected for pheophytin). This resulted in a comparable estimate of 35 mg C m$^{-2}$ y$^{-1}$. In 1976–1978, the annual primary production of microphytobenthos of three stations in the Dollard was between 127 g C m$^{-2}$ y$^{-1}$ and 140 g C m$^{-2}$ y$^{-1}$ [57]. These rates were comparable to those in the western Wadden Sea at that time, where primary production of microphytobenthos increased from 100 g C m$^{-2}$ y$^{-1}$ in 1968 to more than 200 g C m$^{-2}$ y$^{-1}$ in 1981, most probably as the result of eutrophication [29].

The relatively low values in benthic primary production 2018/2019 may be the result of a subsequent reduction in nutrient and carbon input, as was also suggested for causing the observed decline in biomass of benthic fauna in the Dollard from 6 g ADW m$^{-2}$ in the 1970s to 3 g ADW m$^{-2}$ in the 2010s [54]. Based upon a correlation between biomass of benthic fauna (g ADW m$^{-2}$) and primary production (g C m$^{-2}$ y$^{-1}$) as found shallow well-mixed estuarine systems worldwide [99], the primary productivity in the Dollard should have been around 71 g C m$^{-2}$ y$^{-1}$ in the 1970s and 43 g C m$^{-2}$ y$^{-1}$ in the 2010s to support the macrozoobenthic biomass. Although this approach does not take the importance of import into account, it illustrates that nutrient reduction may have resulted in the present findings on the decline in primary production of microphytobenthos.

### Implications for future changes

With respect to seasonal scope for growth of microphytobenthos in temperate turbid environments as our study area, insufficient light only restricts growth the benthic microalgae in midwinter whilst growth may be suboptimal due to photoinhibition in mid-summer (Fig 8). Under the assumption that MPB does not grow when temperatures fall below 0˚C or rise above the maximum temperature tolerance, then growth is restricted in late winter and in late summer. Wind stress was generally high from October to March and low in from April to Sept, whilst rainy days possibly disrupting the microphytobenthic layer occurred occasionally throughout the year (Fig 7 and S8 File).

Our findings suggest that optimal conditions for blooms of microphytobenthos in temperate turbid estuaries occur in February or March, starting after sub-zero temperatures in (late) winter and depending on the occurrence of storms and extreme rainfalls (Figs 7 and 8). Later in spring, both insolation and temperatures become suboptimal for MPB growth, and growth might even become restricted from June to August. When light and temperatures are declining after summer, an autumn bloom might still be possible in August and September, but this cannot be as high as the spring bloom due to the ongoing nitrogen limitation and the onset of the storm season (Figs 7 and 8 and S8 File).

Seasonal dynamics in intertidal environments in north-western Europe are projected to change, e.g., due to a decrease in the frequency of severe winters, an increase in the frequency of heat waves, more variations in salinity due to extreme rainfall events in summer and, possibly, a decrease in ambient light conditions due to sea level rise [100]. Subsequently, biomass and production of MPB is also likely to change, for example by an advanced spring bloom and further growth restrictions during summer, resulting in a shift from unimodal [19] to bimodal seasonality in MPB biomass and production (this paper). These changes will not only lead to changes in spatiotemporal patterns of benthic primary production but also to changes in biodiversity of life under water and ecosystem services including food supply. These changes make it even more urgent to include microphytobenthos biomass and production in monitoring programs.

## Supporting information

**S1 File. Relation daily benthic primary productivity and benthic chlorophyll-a concentrations.** Linear relationships between daily benthic primary productivity (mg C m$^{-2}$ d$^{-1}$) and

corrected benthic chlorophyll-a concentrations (mg m$^{-2}$) for three models with respect to vertical distribution of benthic algae in the top layer of the sediment (n = 6).
(DOCX)

**S2 File. Correlations photosynthetic parameters and environmental conditions.** Correlations between photosynthetic parameters (being $\alpha^\beta$ as the slope of the light-limited part of the curve in mg C (mg CHLa)$^{-1}$ h$^{-1}$ (PAR µE$^{m-2\ s-1}$)$^{-1}$) and P$^\beta_{max}$ as the maximum photosynthetic production rate in mg C (mg CHLa$^{-1}$) h$^{-1}$) and environmental conditions (uncorrected chlorophyll-a, corrected chlorophyll-a and pheophytin-a concentrations (mg m$^{-2}$), median grain size (µm) and mud percentage of the sediment) at the mudflats in the Ems estuary in September 2018, April 2019 and July 2019 (pooled data, n = 6).
(DOCX)

**S3 File. Relation between corrected and uncorrected chlorophyll-a concentrations.** Linear relationships between corrected and uncorrected chlorophyll-a concentrations (mg m$^{-2}$) as determined at the three sampling campaigns.
(DOCX)

**S4 File. Relation hyperspectral sensors and Landsat 7, Landsat 8 and Sentinel 2.** Relationships between NDVI as determined by means of hyperspectral sensors (NDVI_hss; -) during the field surveys and derived from these data for red and near-infrared (NIR) of spectral bands of the Landsat 7 ETM (NDVI_L7;-), the Landsat 8 OLCI (NDVI_L8;-) and the Sentinel 2 (NDVI_S2;-). The different symbols indicate different periods (square: September 2018, circle: April 2019, triangle: July 2019), the diagonal line depicts a 1:1 relationship.
(DOCX)

**S5 File. Names of the Sentinel 2 images (tiles) downloaded.** Sources: (1) DIAS ONDA: https://www.onda-dias.eu/cms/ (Level 1), and (2) Copernicus Open Access Hub: https://scihub.copernicus.eu/dhus (Level 2).
(DOCX)

**S6 File. Overview of cloudless Sentinel 2 images.** Images for the Dollard at which the tidal flats were fully exposed in 2018 and 2019. Astronomical low tide is given for Nieuwe Statenzijl (bordering the Dollard estuary in the south). The timing of the satellite image of 18 September 2018 coincided with that of the 1st sampling cruise and whilst that of 27th July 2019 was taken three days after the 3rd sampling cruise.
(DOCX)

**S7 File. Relations between benthic pigments as a function of NDVI.** Linear relationships between corrected benthic pigment concentrations as a function of NDVI as determined by means of hyperspectral sensors (NDVI_hss; -) and those derived from these data for red and near-infrared (NIR) of spectral bands of the Landsat 7 ETM (NDVI_L7;-), the Landsat 8 OLCI (NDVI_L8;-) and the Sentinel 2 (NDVI_S2;-) during the field surveys in April and July 2019, with benthic pigments as corrected chlorophyll-a concentrations (CHLa_c), the sum of corrected chlorophyll-a and pheophytin-a concentrations (CHPH_c) and as uncorrected chlorophyll-a concentrations (CHLa_u) (n = 20).
(DOCX)

**S8 File. Seasonal dynamics in environmental conditions.** Two sampling stations in 2018, one located within the study area (Groote Gat Noord) and the other one just northwest of this (Bocht van Watum). A) Sea surface temperature (˚ Celcius), with the green line indicating the freezing point of freshwater (0˚C). B) Salinity (psu). C) Concentrations of phosphate (PO$_4^{2-}$;

mg l$^{-1}$. D) Concentrations of ammonia (NH$_4$$^+$; mg l$^{-1}$). E) Concentrations of the sum of nitrite and nitrate (NO$_2$$^{2-}$; + NO$_3$$^{2-}$; mg l$^{-1}$). Data source: RWS. The grey vertical lines indicate the sampling dates of the satellite images, the blue line that of the field survey in 2018.
(DOCX)

## Acknowledgments

During the drafting of the manuscript, Dr. Jacco C. Kromkamp passed away after being ill for some months. We are most grateful for his inspiration and guidance in microphytobenthos research. Besides being an excellent researcher, he was also a much-loved colleague. As Dr. Kromkamp passed away before the submission of the final version of this manuscript, the corresponding author accepts responsibility for the integrity and validity of the data collected and analyzed. We would like to thank the following persons: Charlotte Schmidt (RWS project leader) for her continuous support, Timo Blok, Isabel Brandao, Anne Dekinga, Luc de Monte, Kai Schwalfenberg, Bas Wensveen, Eric Wagemaakers and Evaline van Weerlee (NIOZ) for their assistance with field measurements. Evaline van Weerlee also performed sediment and chlorophyll-a analysis in the laboratory. Karel Bakker (NIOZ) measured DIC concentrations. Sonja van Leeuwen (NIOZ) assisted with modelling of the primary productivity. WaterProof B.V. and the Fieldwork Company provided transport to the tidal flats. We acknowledge the European Commission and the European Space Agency for the Sentinel-2 images. We are most grateful to Peter Herman and Elisabeth Addink as well as to the reviewers Rodney Forster and Dominique Davoult for constructive comments on earlier versions of this manuscript, their input greatly improved our manuscript.

## Author Contributions

**Conceptualization:** Jacco C. Kromkamp, Catharina J. M. Philippart.

**Funding acquisition:** Jacco C. Kromkamp, Catharina J. M. Philippart.

**Investigation:** Pascalle Jacobs, Jaime Pitarch.

**Methodology:** Pascalle Jacobs, Jaime Pitarch, Jacco C. Kromkamp.

**Visualization:** Pascalle Jacobs, Jaime Pitarch, Catharina J. M. Philippart.

**Writing – original draft:** Pascalle Jacobs, Jaime Pitarch, Catharina J. M. Philippart.

**Writing – review & editing:** Pascalle Jacobs, Catharina J. M. Philippart.

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
