## [Decision Letter · Decision Letter 0]

26 Feb 2021

PONE-D-21-01011

Assessing biomass and primary production of microphytobenthos in depositional coastal systems using spectral information

PLOS ONE

Dear Dr. Jacobs,

Thank you for submitting your manuscript to PLOS ONE. After careful consideration, we feel that it has merit but does not fully meet PLOS ONE’s publication criteria as it currently stands. Therefore, we invite you to submit a revised version of the manuscript that addresses the points raised during the review process.

We look forward to receiving your revised manuscript.

Kind regards,

Vona Méléder, Ph.D.

Academic Editor

PLOS ONE

Journal Requirements:

2. In your Methods section, please provide additional location information of the study sites, including geographic coordinates for the data set if available.

5. We note that Figures 1 and 5 in your submission contain map/satellite images which may be copyrighted. All PLOS content is published under the Creative Commons Attribution License (CC BY 4.0), which means that the manuscript, images, and Supporting Information files will be freely available online, and any third party is permitted to access, download, copy, distribute, and use these materials in any way, even commercially, with proper attribution. For these reasons, we cannot publish previously copyrighted maps or satellite images created using proprietary data, such as Google software (Google Maps, Street View, and Earth). For more information, see our copyright guidelines: http://journals.plos.org/plosone/s/licenses-and-copyright.

5.1.    You may seek permission from the original copyright holder of Figures 1 and 5 to publish the content specifically under the CC BY 4.0 license. 

5.2.    If you are unable to obtain permission from the original copyright holder to publish these figures under the CC BY 4.0 license or if the copyright holder’s requirements are incompatible with the CC BY 4.0 license, please either i) remove the figure or ii) supply a replacement figure that complies with the CC BY 4.0 license. Please check copyright information on all replacement figures and update the figure caption with source information. If applicable, please specify in the figure caption text when a figure is similar but not identical to the original image and is therefore for illustrative purposes only.

Reviewers' comments:

Reviewer's Responses to Questions

**Comments to the Author**

1. Is the manuscript technically sound, and do the data support the conclusions?

Reviewer #1: Partly

Reviewer #2: Yes

2. Has the statistical analysis been performed appropriately and rigorously? 

Reviewer #1: Yes

Reviewer #2: Yes

3. Have the authors made all data underlying the findings in their manuscript fully available?

Reviewer #1: No

Reviewer #2: Yes

4. Is the manuscript presented in an intelligible fashion and written in standard English?

Reviewer #1: Yes

Reviewer #2: Yes

5. Review Comments to the Author

Reviewer #1: It is a good paper and brings another solid data set to the calculation of estuarine productivity. Using satellite data adds another dimension to our ability to understand these systems. There are plausible explanations of why biomass (and productivity??) are low in the summer for tidal flats. My feeling is that this is due to grazing being higher than production through the summer.

Please see my comments on the paper marked on the pdf version attached. It requires some changes before acceptance, chiefly the PAR values from the met station are too high. Correction to these will also require correction of the daily primary production as they are used in modelling. I look forward to seeing a revised version, and eventual publication

Rodney Forster

Reviewer #2: The questions discussed in the paper are very interesting but complex. Several previous publications have already discussed these points and concluded about the difficulty to assess biomass and mainly production using spectral information.

The data set presented her is interesting and seems to be of quality.

I consider that the manuscript could deserve publication after addressing some iimportant points.

The authors highlighted important questions but did not really solve them.

First of all, they pointed out an important flaw, that is the fact that satellites measures allow an accurate estimation of Chl.a + Pheophytin-a but not of chl.a alone, that is a major problem to use these data to estimate productivity from chl.a estimates and light availability.

Another flaw is the way the authors measured productivity: they used a lab method, very precise but very far from the actual field productivity. 14C measurements are interesting for physiological considerations and produce potential production as a function of controlled variables but measurements are performed in very artificial conditions, very far from the real life. It can be one of the reaon of the poor correlation between satellite data and estimated productivity. It would be at least necessary to perform measurements on intact sediment samples or, better, to perform in situ measurements of benthic productivity: these two approaches are now quite common and certainly better estimations. The methods used here must be more critically discussed and justified.

Migne et al. (2007, J.Phycol.) tried to do the same kind of approach by using in situ fluorescence measurements to estimate microphytobenthic production. However, in a further paper, the same team (Davoult et al., 2009, Hydrobiologia) demonstrated that the relationship was strongly locality-dependent and should be used with caution despite they obtained a stronger explained variance than you. I believe you should discuss that point.

Some particular points:

p.6-L.128: salinity is now expressed withou unit, like a density

p.16, L. 390: values of the ratio between pheophytin-a and the sum of corrected chl.a and pheophytin-a seem to be wrong and connot be deduced from table 3, please check.

p24,L576: OK but data not from the same year, you should be more cautious because of the high spatial and temporal variability of such results

p25, L619-620: these "optimum" values are questionable (and very high!), it seems better to rely on Ek

p28, L 685-688: I agree with this remark, the same could be apply on the way you measure productivity!

6. PLOS authors have the option to publish the peer review history of their article (what does this mean?). If published, this will include your full peer review and any attached files.

Reviewer #1: **Yes: **Rodney Forster

Reviewer #2: **Yes: **Dominique Davoult

---

## [Author Response · Author response to Decision Letter 0]

13 Apr 2021

Dear editor,

We were pleased to hear that you think our manuscript has merit and we would like to thank you and the two reviewers for their constructive comments on our manuscript “Assessing biomass and primary production of microphytobenthos in depositional coastal systems using spectral information.” [Jacobs et al. PONE-D-21-01011].

Rodney Forster remarked that it is a good manuscript, with a solid data set and that the satellite data adds an extra dimension to help understand coastal systems. His main comment regards the PAR values used. He noted a mistake in the conversion from Watt to microEinstein. We thank the reviewer for addressing this mistake! This has been corrected in this revised version. 

Dominique Davoult thinks that the data set presented is interesting and of quality. He comments that methodological flaws should be discussed in more detail, which we believe we have done this now in this revised version.

A point-by-point reply to the comments of both reviewers can be found in the attached 'Response to reviewers'.

Based on the comments and suggestions made, we think the manuscript has been greatly improved. We appreciated that the reviewers made their identity known to us. We would like to include their names in the acknowledgements but would like to check with you and the reviewers if this is appropriate. 

We hope that you will consider the revised manuscript for publication in PLOS ONE. 

With kind regards,

Pascalle Jacobs on behalf of all authors.

---

## [Decision Letter · Decision Letter 1]

3 Jun 2021

Assessing biomass and primary production of microphytobenthos in depositional coastal systems using spectral information

PONE-D-21-01011R1

Dear Dr. Jacobs,

We’re pleased to inform you that your manuscript has been judged scientifically suitable for publication and will be formally accepted for publication once it meets all outstanding technical requirements.

Kind regards,

Vona Méléder, Ph.D.

Academic Editor

PLOS ONE

Additional Editor Comments (optional):

Reviewers' comments:

Reviewer's Responses to Questions

**Comments to the Author**

1. If the authors have adequately addressed your comments raised in a previous round of review and you feel that this manuscript is now acceptable for publication, you may indicate that here to bypass the “Comments to the Author” section, enter your conflict of interest statement in the “Confidential to Editor” section, and submit your "Accept" recommendation.

Reviewer #1: All comments have been addressed

Reviewer #2: All comments have been addressed

2. Is the manuscript technically sound, and do the data support the conclusions?

Reviewer #1: Yes

Reviewer #2: (No Response)

3. Has the statistical analysis been performed appropriately and rigorously? 

Reviewer #1: Yes

Reviewer #2: (No Response)

4. Have the authors made all data underlying the findings in their manuscript fully available?

Reviewer #1: Yes

Reviewer #2: (No Response)

5. Is the manuscript presented in an intelligible fashion and written in standard English?

Reviewer #1: Yes

Reviewer #2: (No Response)

6. Review Comments to the Author

Reviewer #1: (No Response)

Reviewer #2: (No Response)

7. PLOS authors have the option to publish the peer review history of their article (what does this mean?). If published, this will include your full peer review and any attached files.

Reviewer #1: **Yes: **Rodney Forster

Reviewer #2: **Yes: **Dominique Davoult

---

## [Editor Report · Acceptance letter]

24 Jun 2021

PONE-D-21-01011R1 

Assessing biomass and primary production of microphytobenthos in depositional coastal systems using spectral information 

Dear Dr. Jacobs:

I'm pleased to inform you that your manuscript has been deemed suitable for publication in PLOS ONE. Congratulations! Your manuscript is now with our production department. 

Kind regards, 

on behalf of

Dr. Vona Méléder 

Academic Editor

PLOS ONE